# Long-term droughts may drive drier tropical forests towards increased functional, taxonomic and phylogenetic homogeneity

Jesús Aguirre-Gutiérrez [1,2 ✉], Yadvinder Malhi [1], Simon L. Lewis [3,4], Sophie Fauset [5], Stephen Adu-Bredu[6], Kofi Affum-Baffoe[7], Timothy R. Baker[3], Agne Gvozdevaite[1], Wannes Hubau [3,8], Sam Moore[1], Theresa Peprah[6], Kasia Ziemińska[9,10], Oliver L. Phillips [3] & Imma Oliveras[1]

Tropical ecosystems adapted to high water availability may be highly impacted by climatic changes that increase soil and atmospheric moisture deficits. Many tropical regions are experiencing significant changes in climatic conditions, which may induce strong shifts in taxonomic, functional and phylogenetic diversity of forest communities. However, it remains unclear if and to what extent tropical forests are shifting in these facets of diversity along climatic gradients in response to climate change. Here, we show that changes in climate affected all three facets of diversity in West Africa in recent decades. Taxonomic and functional diversity increased in wetter forests but tended to decrease in forests with drier climate. Phylogenetic diversity showed a large decrease along a wet-dry climatic gradient. Notably, we find that all three facets of diversity tended to be higher in wetter forests. Drier forests showed functional, taxonomic and phylogenetic homogenization. Understanding how different facets of diversity respond to a changing environment across climatic gradients is essential for effective long-term conservation of tropical forest ecosystems.

[1] Environmental Change Institute, School of Geography and the Environment, University of Oxford, Oxford, UK. [2] Biodiversity Dynamics, Naturalis Biodiversity Center, Leiden, The Netherlands. [3] Ecology and Global Change, School of Geography, University of Leeds, Leeds, West Yorkshire, UK. [4] Department of Geography, University College London, London, UK. [5] School of Geography, Earth and Environmental Science, University of Plymouth, Plymouth, UK. [6] CSIR-Forestry Research Institute of Ghana, University Post Office, KNUST, Kumasi, Ghana. [7] Mensuration Unit, Forestry Commission of Ghana, Kumasi, Ghana. [8] Service of Wood Biology, Royal Museum for Central Africa, Tervuren, Belgium. [9] Arnold Arboretum of Harvard University, Boston, MA, USA. [10]Present address: Department of Plant Ecology and Evolution, Uppsala University, Uppsala, Sweden. ✉email: jesus.aguirregutierrez@ouce.ox.ac.uk

The biosphere is experiencing unprecedented changes in biodiversity and restructuring of species composition at local and global scales[1]. Evidence gathered from the Intergovernmental Science-Policy Platform on Biodiversity and Ecosystem Services (IPBES) demonstrates large biodiversity declines, with around one million species threatened with extinction, posing a threat to the functioning of ecosystems and human well-being[2]. Some of the main drivers of such biodiversity decline are climate related: altered precipitation and temperature patterns, and extreme weather events[3]. In West Africa, a drying environment over the last decades has been associated with changes in forest composition, leaf phenology and community-level functional traits[4,5], i.e. the intrinsic morphological/physiological characteristics of species. Future changes in climatic conditions may not only impact forest taxonomic diversity and functional trait composition but even threaten entire phylogenetic clades of forest ecosystems[6]. Aside from climatic conditions, soil characteristics, e.g., texture and fertility, may also determine forest responses to a changing climate. For instance, forest soils high in clay may be able to maintain higher water availability over longer periods during droughts than sandy soils where the water holding capacity tends to be lower[7]. Moreover, tropical forests in drier regions tend to be associated with nutrient richer soils in comparison to wetter tropical forests, which may confer further resistance to a changing climate[8]. Such soil–rainfall–plant feedbacks may be disrupted under a drying climate, especially in nutrient poor soils and thus strongly affect the functioning of forest ecosystems.

Although much work has been done focusing on species richness distribution patterns[9], the importance of other facets of diversity, such as functional diversity[10,11] and phylogenetic composition[12] have been increasingly highlighted. It has become evident the role that high functional and phylogenetic diversity levels may play for increasing the ecosystems resilience to changes in environmental conditions. Functional diversity can enhance the capacity of ecosystems to resist or be resilient to new environmental conditions[13,14] and could prevent them from shifting into alternative states, e.g., shifts from a closed-canopy tropical forest to an open savanna-like vegetation or vice versa[15]. Phylogenetic diversity can render insights about the species evolutionary history, adaptations to past environmental conditions and into their irreplaceability in a community[16]. These three facets of diversity, taxonomic, functional and phylogenetic, can contribute to ecosystem stability and functions such as carbon sequestration[17], water capture[18] and buffering of temperature variability[19], and therefore, decreases in any of them could potentially generate negative forest feedbacks and disturb the functioning of ecosystems.

Although there is compelling evidence that tropical forest communities are responding to atmospheric change[20] and that as a result such communities may experience strong species declines in the near future[21], we are just beginning to understand how such forests respond to a shifting environment on multidecadal time spans[22]. Moreover, the question remains as whether tropical forests along climatic gradients show coordinated responses to climate change regarding their functional, taxonomic and phylogenetic facets of diversity. It has been recently shown that the plant traits composition in West African tropical forests has shifted more in drier than wetter forests, arguably as a result of a changing climate[4]. In such drier forests the abundance of deciduous species is increasing, which could be generating forest communities better adapted to a drying climate[5]. Esquivel-Muelbert et al.[22] suggest that in Amazonia, drought adapted species may be expanding their range and increasing in abundance, and in SE Asian forests there is evidence of shifts in composition[23] and carbon sink dynamics after extreme weather events, such as El Niño, but without a uniform response along disturbance gradients[24]. Overall, it is not yet understood if such possible shifts in functional, taxonomic and phylogenetic diversity along climatic gradients and in response to a changing climate are taking place, if such shifts are in the same direction (i.e., increases or decreases in diversity) and if so with what intensity. Understanding the above-mentioned processes and filling this knowledge gap is of relevance as changes in the three facets of diversity may have different implications for the functioning of ecosystems and their responses to environmental changes[25].

Here, we investigate if and how climate change, given an observed multidecadal drying trend[5], has affected the functional, taxonomic and phylogenetic diversity of tropical forests in West Africa, and if the forests responses to climate change are dependent on their position along the climatic gradient. We specifically ask (1) if and to what extent there have been shifts in the three facets of diversity across time; (2) to what extent such shifts are explained by changes in climate? and 3) if the diversity shifts are synchronous and monodirectional, i.e., whether diversity uniformly increases or declines across the climatic gradient. We expect that a drying trend would be reflected in overall diversity decreases along the water deficit gradient, however, forest communities located in the drier end of the water deficit gradient may experience higher climatic stress and therefore the diversity changes may be stronger in those locations. Responses in the three facets of diversity may be determined by soil characteristics in addition to climatic conditions; for example, soils rich in nutrients and with higher water holding capacity (e.g., higher clay content) may buffer drought impacts on forest communities[7], as drought resilience may vary not only with depth to water table but also with soil nutrient content[26]. Therefore, we also investigate the role of soil characteristics on the response of the three facets of diversity along the climatic gradient and across time.

We analyse changes in functional (FDis)[27] (Supplementary Table 2), taxonomic (Simpson diversity index)[28,29], and phylogenetic diversity (mean pairwise phylogenetic distance, MPD)[30] of 21 unique vegetation plots from the African Tropical Rainforest Observation Network (AfriTON; Fig. 1) across time (range 1987–2013; Supplementary Table 1). To assess shifts in the three facets of diversity we use their yearly rate of change and apply Bayesian estimation[31,32]. To investigate the role that climate may play on determining changes in the three facets of diversity we calculate the mean maximum climatic water deficit and vapour pressure deficit for the full term of the study ($MCWD_{Full}$ and $VPD_{Full}$ respectively), for each census time and calculate the absolute changes for each metric ($\Delta MCWD_{Abs}$ and $\Delta VPD_{Abs}$). Then we conduct a principal component analysis of the soil characteristics (Supplementary Table 3). We construct different statistical models under a Bayesian framework (Supplementary Table 4) to test for the effects of climatic and soil conditions on the changes in diversity.

We find differential responses of tropical forests to a changing environment with direr tropical forests showing stronger declines in the three facets of diversity in contrast to wetter tropical forests. Our results fill knowledge gaps on the coordination of changes in biodiversity in tropical forest as a response to climate changes, and on the extent to which forest communities may be susceptible to a changing environment depending on their current position along the climatic gradient.

## Results

### Changes in functional, taxonomic and phylogenetic diversity.
Overall, our results show that the three facets of diversity changed across time, and that such changes were not necessarily

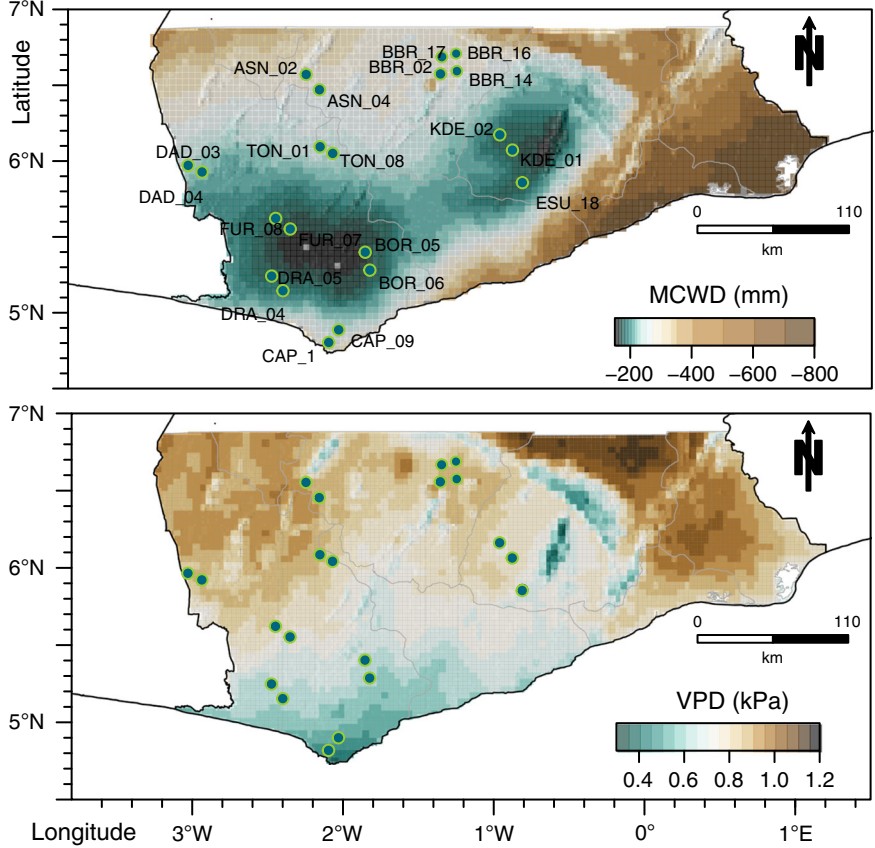

**Fig. 1 The distribution of vegetation plots (green dots) in Ghana, West Africa.** The top panel shows the maximum climatic water deficit (MCWD) and the bottom the vapour pressure deficit (VPD) over the study area averaged over the full study period. The plot data from the African Tropical Rainforest Observation Network (AfriTON) dataset[20,63] and the climatic data were obtained using the TerraClimate dataset[80]. See also Supplementary Table 1 for full plot details.

synchronous and equal across the climatic gradient, with the three facets of diversity usually decreasing more at the drier end than at the wetter end of the water availability spectrum (Fig. 2a–c and Supplementary Fig. 1).

Forest communities in drier sites (more negative maximum climatic water deficit, MCWD) experienced on average stronger declines (Probability = 95.2%) in functional diversity (FDis) across time than forest communities in wetter areas (Fig. 2a, d). Taxonomic diversity (Simpson) changes across time differed also along the climatic gradient (Fig. 2b), with drier forest communities showing on average stronger declines (Probability = 96.9.%) in taxonomic diversity than communities located in wetter areas (Fig. 2e). The phylogenetic diversity (MPD) showed large average decreases along the climatic gradient (Fig. 2c), with forests at the drier end of the water deficit spectrum showing on average larger ($\mu = -0.03$) but not statistically different rates of phylogenetic diversity declines than forests in wetter locations (Probability = 61.7%; Fig. 2f). In general, the forest communities have transitioned towards lower phylogenetic diversity across time (Fig. 2d–f). The phylogenetic and functional diversity changes were not significantly correlated (Supplementary Fig. 2) even though all traits that conform the functional diversity metric (FDis), showed significant phylogenetic signal (Supplementary Table 5).

In summary, the drier forests are transitioning towards increasingly more homogenous forest communities, diverging further from wetter forests in functional, taxonomic and phylogenetic diversity. The changes in the three facets of diversity do not appear to be driven by changes in the plots' basal area (see

extended community dynamics text in SI) as the changes in basal area were not related to changes in functional ($R^2 = -0.01$, $P = 0.94$), taxonomic ($R^2 = 0.28$, $P = 0.21$) or phylogenetic diversity ($R^2 = -0.17$, $P = 0.46$). Moreover, the species with strongest changes in basal area (Supplementary Fig. 3) did not show phylogenetic clustering as they did not cluster in specific locations of the phylogenetic tree (Supplementary Fig. 4).

**Climatic and soil drivers of changes in facets of diversity.** The full-term (1964–2013) water deficit ($MCWD_{Full}$) ranged between $-167.36$ and $-300.74$ mm along the climatic gradient, the MCWD became more negative across all forest plots over the study period and increased their VPD (Supplementary Table 1). Soil properties varied greatly among forest communities with some of the main soil properties such as cation exchange capacity ranging between 10.99 and 29.14 mmol kg and soil phosphorous ranging between 34.97 and 137.75 mg kg (Supplementary Table 3). Climatic and soil conditions partly explained the changes in functional, taxonomic and phylogenetic diversity that have occurred over the past three decades in forest communities in West African tropical forests (Table 1). The best statistical models (Table 1; Supplementary Table 6) showed that while changes in functional and taxonomic diversity were best explained by changes in climatic conditions ($\Delta MCWD_{Abs}$), changes in phylogenetic diversity were also strongly mediated by soil characteristics (Table 1). Results for the second best models for the three facets of diversity following the leave one out cross-validation are also shown in Supplementary Table S7. Functional

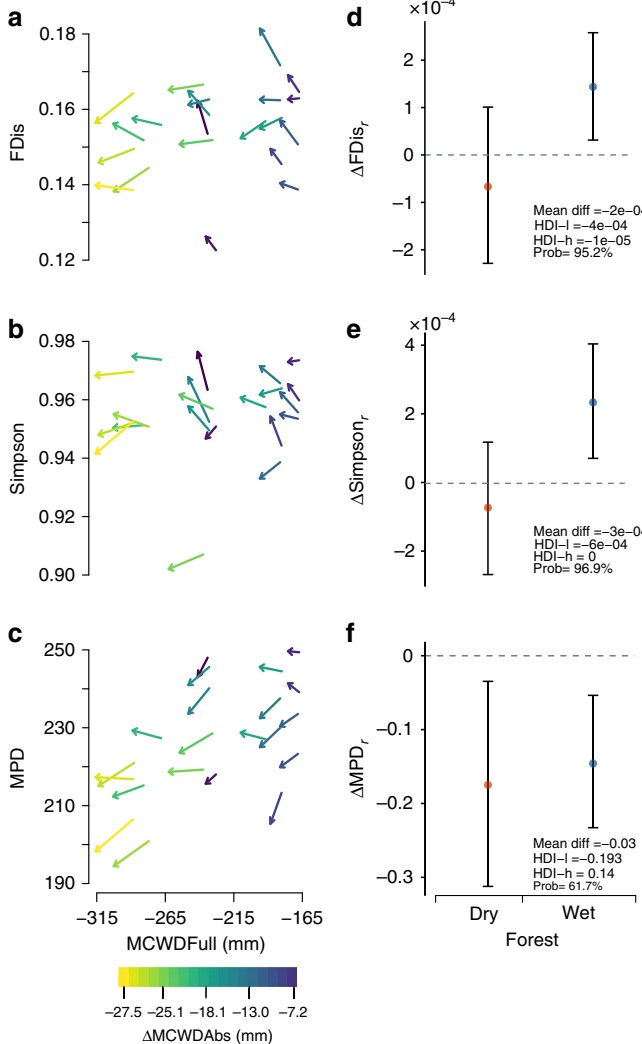

**Fig. 2 Changes in the three facets of diversity, functional (FDis), taxonomic (Simpson) and phylogenetic (MPD) across time and climatic gradient.** In **a–c** each arrow represents a vegetation plot (*n* = 21), with the tail of the arrow showing the diversity level during the first vegetation census and the head of the arrow the diversity level during the last vegetation census. The slope of the arrow represents the change in diversity across time, so that arrows pointing upwards show increases and downwards decreases in diversity. Arrow colours reflect the absolute change in maximum climatic water deficit ($\Delta MCWD_{Abs}$) experienced by the forest community (colour bar at the bottom of the figure). The *X*-axis shows the full-term maximum climatic water deficit (MCWD$_{Full}$ covering the 1964–2013 period) and the *Y*-axis the facets of diversity. **d–f** Show the average rate of change (coloured dots) and highest density intervals (vertical lines) in the three facets of diversity ($\Delta FDis_r$, $\Delta Simpson_r$ and $\Delta MPD_r$) after grouping the vegetation plots as belonging to dry or wet forest (*n* = 21). The horizontal dotted line represents no change in diversity with positive values showing increases and negatives values decreases in diversity. The insets on the bottom right corner show the average difference in diversity change between dry and wet forests. Negative average difference values depict a stronger loss in the diversity facet in drier forests in comparison to wetter forests. The posterior highest density intervals (HDI-l: lower; HDI-u: upper) and probability change (Prob) values are also shown.

diversity decreased the most (up to $-3.9^{e-4}$ yearly rate and $-7.2^{e-3}$ in total for plot BBR_16) in areas that experienced the strongest negative $\Delta MCWD_{Abs}$ ($-27.5$ mm; Fig. 3a) and increased (yearly rate of up to $4.3^{e-4}$ for CAP_10 and $4.2^{e-4}$ for

KDE_02) in areas that experienced the smallest $\Delta MCWD_{Abs}$ ($-7.5$ mm; $R^2_{adj} = 0.41$; Table 1). Taxonomic diversity tended to increase in areas where $\Delta MCWD_{Abs}$ was small and decreased in areas where $\Delta MCWD_{Abs}$ was strongly negative ($R^2_{adj} = 0.24$; Fig. 3b; Table 1). Changes in phylogenetic diversity (MPD) were best explained by the soil components related to the three PC axes and their interaction with climatic conditions (Table 1; $R^2_{adj} = 0.9$). PC1 is a nutrients axes (eCEC), PC2 is an acidity-calcium axis (pH-Ca) and PC3 a soil texture axis (%Sand and %Clay; Supplementary Table 3). Overall, most forest communities (15 vegetation plots) decreased in phylogenetic diversity ($\Delta MPD_r$ up to $-0.47$ for plot ESU_18). Communities where the water deficits increased (became drier; up to $-27.5$ mm) showed the stronger declines in phylogenetic diversity as soils became more acidic and in more sandy soils (Fig. 3c, d). Forest communities found in areas with smaller water deficits but average or higher than average soil acidity and in more sandy soil tended to show stronger declines in phylogenetic diversity in contrast to communities with higher water deficits (Fig. 3e, f). Forest communities in areas with high soil nutrients or in more sandy soils and which experienced small increases in VPD (mostly drier communities), showed stronger MPD declines in comparison to communities that experienced the strongest changes in VPD (overall wetter communities; Fig. 3g, h).

## Discussion

Here, we show differential shifts in functional, taxonomic and phylogenetic diversity in tropical forest communities distributed along a strong climatic gradient and through decadal time spans in West Africa. These shifts are partly explained by changes in climatic conditions and by inherent soil properties. Our findings show that forests that normally experience higher seasonal water deficit, and that became drier through time, tended to become more homogeneous in the three facets of diversity under a drying climate. In contrast, wetter forests showed on average increases in functional and taxonomic diversity under a drying climate. Thus, our results suggest that drier tropical forests that have experienced increases in water deficits may be less resistant (in terms of community composition) to a drying environment than wetter forest communities.

The observed shifts in facets of diversity across tropical forest communities provide a fundamental advance in our understanding of how forests may respond under a drying climate, showing that such responses may depend on the forest communities position along the climatic gradient and the changes in water availability experienced across time. This advances from previous evidence of changes in plant composition towards drought tolerant species[5] and trait compositional changes in West African tropical forests[4] by bridging the functional, taxonomic and phylogenetic diversity responses. Although we did not investigate how changes in each of the three facets of diversity affect ecosystem functioning, there is strong support from recent studies showing how decreasing functional[33], taxonomic[34,35] and/ or phylogenetic[36,37] diversity may cause severe loss of forest functions, such as resources uptake, cycling and biomass production and resilience to a changing climate across spatial and temporal scales[25]. As such, ecosystem functions of communities that show decreases across all three facets of diversity could be especially vulnerable under a drying climate.

Our results partly meet our expectation of decreasing diversity given a drying trend. Such changes in diversity were not equal along the climatic gradient and depended on the climatic water deficit, its change and the change in the VPD experienced by the forest communities. In general communities in drier locations also experienced stronger declines in water availability

**Table 1 Linear regression result for the most parsimonious models carried out under a Bayesian framework explaining the functional (FDis), taxonomic (Simpson) and phylogenetic (MPD) rates of diversity changes as a function of climatic and soil drivers.**

| Metric | Parameter | Median | HDI (l) 50% | HDI (h) | HDI (l) 89% | HDI (h) | HDI (l) 95% | HDI (h) | ROPE | Rhat |
|---|---|---|---|---|---|---|---|---|---|---|
| $\Delta FDis_r$ | Intercept | 6.34E−05 | 3.71E−05 | 9.35E−05 | −1.69E−06 | 1.38E−04 | −2.11E−05 | 1.54E−04 | 0.13 | 1.00 |
| | $\Delta MCWD_{Abs}$ | 1.40E−04 | 1.15E−04 | 1.73E−04 | 6.59E−05 | 2.08E−04 | 5.44E−05 | 2.35E−04 | 0.00 | 1.01 |
| | Plot area | −6.28E−05 | −9.37E−05 | −3.37E−05 | −1.42E−04 | 8.89E−06 | −1.63E−04 | 3.00E−05 | 0.15 | 1.00 |
| $\Delta Simpson_r$ | Intercept | 1.19E−04 | 7.19E−05 | 1.65E−04 | 2.92E−06 | 2.28E−04 | 2.35E−05 | 2.62E−04 | 0.08 | 1.00 |
| | $\Delta MCWD_{Abs}$ | 1.61E−04 | 1.02E−04 | 1.99E−04 | 3.58E−05 | 2.80E−04 | 8.16E−06 | 3.12E−04 | 0.00 | 1.00 |
| | Plot area | −8.15E−06 | −5.23E−05 | 4.24E−05 | −1.22E−04 | 1.04E−04 | 1.46E−06 | 1.30E−04 | 0.41 | 1.00 |
| $\Delta MPD_r$ | Intercept | −0.17 | −2.03E−01 | −1.50E−01 | −0.24 | −9.77E−02 | −2.56E−01 | −7.02E−02 | 0.00 | 1.00 |
| | PC1 | −0.06 | −6.71E−02 | −4.32E−02 | −0.09 | −2.27E−02 | −9.81E−02 | −1.02E−02 | 0.00 | 1.00 |
| | PC2 | −0.04 | −4.78E−02 | −2.59E−02 | −0.07 | −5.29E−03 | −7.98E−02 | 4.51E−03 | 0.10 | 1.00 |
| | PC3 | 0.01 | −7.65E−03 | 2.94E−02 | −0.03 | 6.34E−02 | −4.66E−02 | 8.42E−02 | 0.51 | 1.00 |
| | $\Delta VPD_{Abs}$ | 0.20 | 1.76E−01 | 2.28E−01 | 0.13 | 2.67E−01 | 1.01E−01 | 2.84E−01 | 0.00 | 1.00 |
| | $\Delta MCWD_{Full}$ | −0.12 | −1.63E−01 | −9.79E−02 | −0.21 | −3.66E−02 | −2.26E−01 | −4.11E−03 | 0.00 | 1.00 |
| | $\Delta MCWD_{Abs}$ | −0.03 | −5.09E−02 | −1.11E−03 | −0.09 | 3.91E−02 | −1.11E−01 | 5.91E−02 | 0.31 | 1.00 |
| | Plot area | 0.06 | 4.35E−02 | 6.76E−02 | 0.02 | 8.74E−02 | 1.90E−02 | 1.07E−01 | 0.00 | 1.00 |
| | PC1: $\Delta VPD_{Abs}$ | 0.06 | 3.93E−02 | 8.23E−02 | 0.00 | 1.11E−01 | −1.59E−02 | 1.32E−01 | 0.08 | 1.00 |
| | PC2: $\Delta VPD_{Abs}$ | −0.02 | −2.63E−02 | −4.08E−03 | −0.04 | 1.36E−02 | −5.61E−02 | 2.05E−02 | 0.57 | 1.00 |
| | PC3: $\Delta VPD_{Abs}$ | 0.15 | 1.32E−01 | 1.84E−01 | 0.07 | 2.12E−01 | 4.82E−02 | 2.34E−01 | 0.00 | 1.00 |
| | PC1: $\Delta MCWD_{Full}$ | −0.06 | −9.72E−02 | −3.46E−02 | −0.15 | 2.23E−02 | −1.58E−01 | 5.86E−02 | 0.12 | 1.00 |
| | PC2: $\Delta MCWD_{Full}$ | −0.05 | −6.26E−02 | −3.45E−02 | −0.08 | −1.02E−02 | −9.25E−02 | 2.29E−03 | 0.05 | 1.00 |
| | PC3: $\Delta MCWD_{Full}$ | −0.17 | −1.97E−01 | −1.46E−01 | −0.24 | −1.05E−01 | −2.50E−01 | −7.43E−02 | 0.00 | 1.00 |
| | PC1: $\Delta MCWD_{Abs}$ | 0.03 | 1.71E−02 | 4.76E−02 | −0.01 | 7.57E−02 | −2.69E−02 | 8.32E−02 | 0.23 | 1.00 |
| | PC2: $\Delta MCWD_{Abs}$ | 0.08 | 6.28E−02 | 9.40E−02 | 0.04 | 1.17E−01 | 2.16E−02 | 1.29E−01 | 0.00 | 1.00 |
| | PC3: $\Delta MCWD_{Abs}$ | 0.11 | 8.78E−02 | 1.28E−01 | 0.05 | 1.56E−01 | 3.54E−02 | 1.78E−01 | 0.00 | 1.00 |

Several different models were fitted (see Supplementary Tables S4 and S6) to investigate the drivers of changes of each diversity facet. The most parsimonious model, shown above, was selected based on the leave one out cross-validation information criterion (LOOIC) and expected log predicted density (ELPD). Only the most statistically important interactions (lowest ROPE values, i.e., <0.10) are shown in Fig. 3. *Rhat* potential scale reduction statistic. *HDI* highest density interval, *l* low, *h* high, *ROPE* region of practical equivalence to test the importance of parameters with values of 0 or close to 0 reporting a more significant effect.

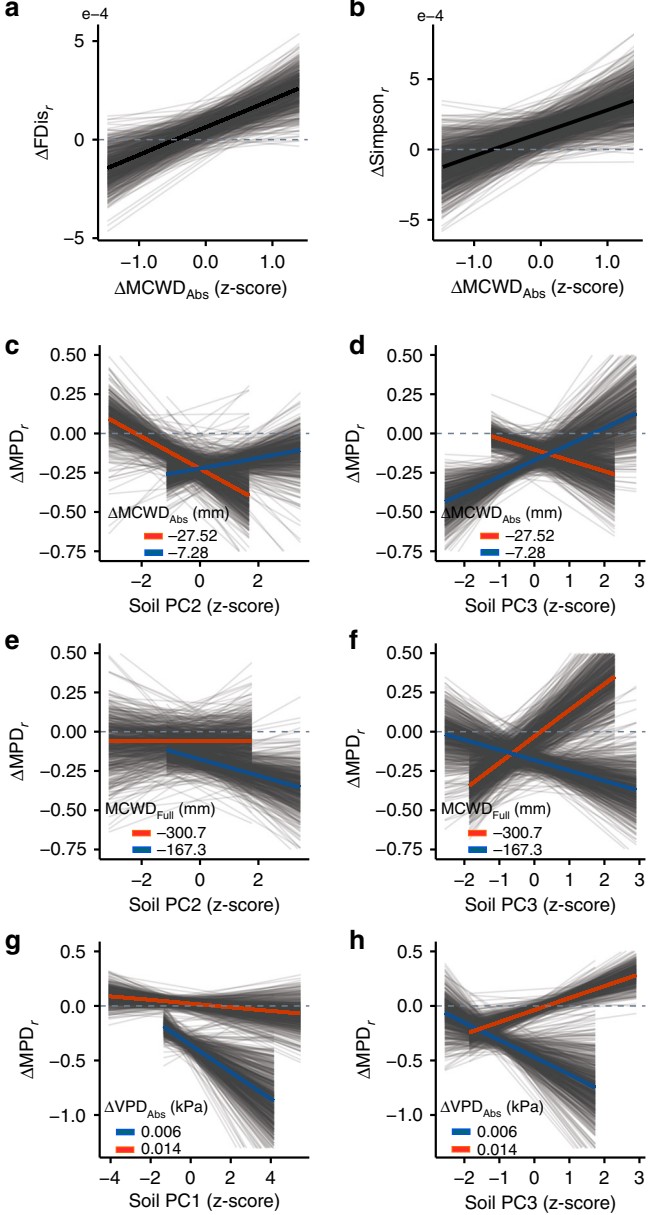

**Fig. 3 Climatic and soil drivers of observed rates of change in the three facets of diversity.** **a** Functional ($\Delta$FDis$_r$), **b** taxonomic ($\Delta$Simpson$_r$) and **c–h** phylogenetic ($\Delta$MPD$_r$) diversity in West African forest communities. Changes in functional and taxonomic diversity were mainly explained by the absolute changes in the maximum climatic water deficit ($\Delta$MCWD$_{Abs}$). Observed changes in phylogenetic diversity were better explained by the soil characteristics covered by the three PC axes (Supplementary Table 3) and their interaction with climatic drivers ($\Delta$MCWD$_{Abs}$, $\Delta$MCWD$_{Full}$, $\Delta$VPD$_{Abs}$). PC1: eCEC(+), magnesium(+) and nitrogen(+); PC2: pH(−), Fe(+) and Ca (−); PC3: %Clay(−) and %Sand(+). The solid black fitted line shows the mean posterior prediction for the functional and taxonym diversity change models. The red and blue fitted lines shows the mean posterior predictions for the phylogenetic diversity based on the minimum and maximum values of the climatic drivers included in the model (Table 1). Grey lines show 700 random draws from the posterior distribution representing variability of the expected model fit. $n = 21$ unique vegetation plots.

(see Fig. 2a). We expected drier forests that further experienced strong negative changes in water availability to suffer more from such drying conditions than wetter forests given they may already be at their climatic threshold. It is possible the forest communities

that have experienced high water deficits for long periods of time and that experienced further negative changes in water availability to become less diverse after prolonged droughts (see[14]) such as those experienced in West Africa. Moreover, we show how even slight increases in VPD may cause diversity to decline in areas already under water shortages, as is the case of the drier tropical forest in Ghana (Fig. 3g, h). Increased VPD may lead to greater transpiration and lower photosynthetic activity specially under drought conditions. Yuan et al.[38] have shown how increases in VPD can reduce vegetation growth as a result of changes in photosynthetic activity, and may also cause faster mortality during drought for tree seedlings[39], which could thus affect ecosystem functioning. Our results and other recent work analysing functional trait shifts in tropical forests[4] evidence how climate may be acting on the filtering of species in such high water deficit communities, which are already under high climatic pressure and at the edge of their climatic suitability. We show that those communities are in general becoming functionally, taxonomically and phylogenetically more homogeneous than forest communities in areas less restricted by water availability, and thus they may be less resistant[40] to further changes in climatic conditions. A recent study for West African tropical forests shows that single plant traits at the community level are shifting the most in drier forests[4], which supports our findings of such forests being the ones also changing the most in their overall diversity. In Neotropical forests Esquivel-Muelbert et al.[22] found increases in dry-affiliated taxa suggesting those tropical forests are also shifting and favouring communities that are better adapted to drier conditions. It is possible that the forest communities in the drier end of the forests we analysed have reached such threshold because of the long-term drying trend experienced in this region[5]. This ultimately could create a less diverse tropical forest that may in principle cope better with a drying climate[14]. However, such changes in plant community composition may also impact the way the ecosystem functions and its contributions to people, as for biomass production, carbon capture and biogeochemical cycling[41].

Soil properties are a main determinant of species distributions and ecosystem functioning[7] and provide vital nutrients to plants[42]. Furthermore, depending on the soils water holding capacity they act as water reservoir for plants through the dry seasons or during extreme weather events such as El Nino[43]. The ecosystem functions carried out in tropical forests are the result of not only the species available in the community but more specifically of their functional traits and the inherent phylogenetic relationships between them[44–46]. Our results show that soil nutrient content, acidity and texture strongly determined the observed changes in phylogenetic diversity over time in West African forests. Our results reveal that forest communities in nutrient poorer, sandy and acidic soils are show to be the ones displaying slight increases in phylogenetic diversity under drying, response that is also mediated by the climatic conditions in the forest community (see Fig. 3c–h). Such communities resemble the areas of climate refugia of tropical forests in Ghana discussed by Maley[47] (see Fig. 5 in[47]) and such areas may also encompass the soil characteristics and higher levels of phylogenetic diversity found on the wetter tropical forests shown in this study. Hall and Swaine[48] have shown that Ghanaian drier tropical forests tend to be richer in soil nutrients than wetter forests and Meir and Pennington[8] suggested the same for drier in comparison to wetter neotropical forests, which concurs with our findings, as the few communities increasing in phylogenetic diversity occurred on soils with lower than average nutrients content (see Fig. 3c). This association of poor soils and high phylogenetic diversity could in principle also determine the tree community composition, and thus the phylogenetic associations between the species present in

the community due to a strong soil–plant feedback that is phylogenetically dependent[49]. Such soil–plant feedback may be disrupted under a drying climate thereof impacting the local soil–plant symbiotic interactions and pathogen communities[36], and likely having a negative effect on the plant communities and general forest ecosystem. The best model explaining changes in functional and taxonomic diversity did not include any of the soil related drivers, although it has been shown for other tropical forests, such as the Amazon (e.g.[50,51]), that soil is a main driver of species distributions[7,52]. Soil fertility has also been suggested to be one of the main determinants of plant species distributions in Ghanaian tropical forests[53], however, we did not find evidence that it also mediates their response to a changing climate by shifting their functional or taxonomic diversity.

We observed an asynchronous shift in the facets of diversity along the climatic gradient, which suggests that communities respond in different ways to environmental changes depending on their current position along the gradient (e.g., if in wetter or drier locations). Such response may be mediated by functional trait characteristics, which underpin the capacity of communities to pose an effect on the environment and also respond to climate changes. Our results suggest that drier forest communities are changing their functional trait composition in part as a response to a drying climate. Such changes are selecting for species better adapted to drier conditions as shown by the already observed increases in the abundance of deciduous species, with lower leaf area: sapwood area ratios (LA:SA) and higher photosynthetic capacity in West African tropical forests[5]. In contrast, wetter forest communities, which in general experienced smaller changes in water availability across time, do not show such species filtering patterns, probably given their higher atmospheric and ground water availability in comparison to that available for drier forests. In Bolivian Neotropical rainforests, Toledo et al.[54] have shown how species richness and the probability of species occurrence is greatly determined by climatic conditions, especially rainfall. Moreover, Esquivel-Muelbert et al.[55] showed that water availability is a main driver of diversity of tree species in the Neotropics, suggesting that the distribution of many tree taxa is physiologically limited by the moisture gradient. Our findings show a clear effect of climatic position on the three facets of diversity, and in addition, we find that wetter forest communities have higher diversity levels and have been less impacted by changes in moisture conditions than drier forest communities. African forests have experienced periods of pronounced drought during the late twentieth century[56] and climate variability over the Holocene, which may have led them to developed a historical acclimation to drought and may explain their more pronounced resilience to current drying trends as compared with the less climatically variable Amazonian forests. Central and West African forests have been more severely disturbed by large-scale climatic anomalies throughout the Pleistocene and the Holocene than other tropical forest regions[57]. There is scientific evidence of abrupt changes in climatic conditions and extreme droughts occurring around 4000 cal year BP across all West and northern Africa, with important implications for atmospheric dynamics across Africa[58]. Such abrupt changes in climatic conditions caused the expansions of open forest, savanna woodland and grasslands and the contraction of rainforest in West Africa with only subsequent development of regenerating forests[59]. As forest recovery is a slow process that can go on for several centuries[60], these relatively recent cycles of drought and forest disturbance may have had a legacy and could be partly determining the patterns of distribution of functional, taxonomic and phylogenetic diversity observed.

In sum, we demonstrate that tropical forests in West Africa show changes in their facets of diversity partly due to a changing climate and partly to their dependence on intrinsic soil properties. Drier forest communities that have experienced stronger decreases in water availability have undergone functional, taxonomic and phylogenetic diversity homogenisation. Such homogenisation process could have negative effects on the current functioning of such tropical ecosystems and therefore on their contribution to people's livelihoods[61].

## Methods

**Study area and vegetation census**. We focus on the forest zone of Ghana, West Africa, which ranges in rainfall from 2000 mm near the southwest coast to around 700 mm near the forest-savanna transition[5,62]. We obtained vegetation census data from 28 permanent vegetation plots that are part of the African Tropical Forest Observation Network (AfriTRON; www.afritron.org)[63]. The plots were obtained from the ForestPlots.net database (www.forestplots.net)[64]. The plots were originally established by the Forestry Commission of Ghana, which also collected most of the first vegetation census data, as part of the long-term forest monitoring programme[65]. Most species identifications were carried out by Hawthorne[66]. We chose vegetation plots that were measured at least twice, with at least 10 years difference between the first (from 1980s or early 1990s, first time period—T1) and second census (2010–2013, second time period—T2). All vegetation plots had an original size of 1 ha but some of them experienced logging in small portions of their area after the first census, therefore the disturbed subplots were excluded from the analysis in both time periods (Supplementary Table 1). The size difference between plots was accounted in our analysis (see statistical analysis section) and such plots did not show a different response pattern than un-logged ones. Seven out of the 28 plots experienced anthropogenic fire events and were thus excluded from the analysis, leading to a total of 21 unique plots used (Fig. 1). The vegetation plots are distributed across the forest zone encompassing varied climatic conditions: in general, plots further north towards the forest-savanna transition experience higher water and VPDs than those in the centre and south of the study area. The study area has experienced variation in climatic conditions over the last century, with a strong drying trend and several drought events between the 1970s and 2005[5]. In each plot, all individuals with a diameter at breast height (DBH) ≥ 10 cm, were measured and identified to the species (94% period 1 and 93.5% period 2) or genus level (6.0% and 6.5%, respectively) ($n = 11,110$ individuals in period 1 from 347 different taxa and 11,309 individuals in period 2 from 350 taxa in period 2). Detailed information on the collection, quality and validation of the vegetation inventories is available in www.forestplots.net and in the AfriTRON site www.afritron.org.

**Functional diversity calculation**. We calculated functional trait diversity (FDis) using in situ collected plant functional traits that are hypothesised to be of importance for tropical forests to adapt or respond to a drying climate (Supplementary Table 2). We collected plant functional traits during 2015 and 2016 in Ghana as part of the Global Ecosystems Monitoring TRAIT network campaign (GEM; www.gem.tropicalforests.ox.ac.uk), named KWAEEMA. The traits collection was carried out at seven different 1 ha plots across the climatic gradient of the study area. The trait sampling plots were located in the humid forest zone in Ankasa National Park (two plots of 1 ha each; latitude: 5.267, longitude: −2.693; 5.2710–2.692), in the semi-deciduous forest zone in Bobiri (two plots of 1 ha each; 6.691, −1.338; 6.704, −1.318) and on the dry forest zone in Kogyae Strict Wildlife Reserve (three plots of 1 ha each; 7.261, −1.150; 7.302, −1.180; 7.301, −1.164). For further details on site characteristics, plot biomass, productivity and carbon cycling of these plots see[67]. The following traits from the leaf, hydraulics and wood economics spectrum were collected: LA:SA, potential stem specific conductivity (kp), vessel lumen fraction (VLF), vessels diameter (VD), vessel density (pV), leaf area (Area$_L$), specific leaf area (SLA), leaf nitrogen(N$_L$) and phosphorus (P$_L$) content, leaf thickness (Thickness$_L$), photosynthetic capacity at maximum carbon assimilation rates (A$_{max}$) and at light saturated carbon assimilation rates (A$_{sat}$), adult maximum height (Height$_{max}$), wood density (WD), phenology, guild and nitrogen fixing capacity (Supplementary Table 2). For a fuller description on the field trait sampling see Oliveras et al.[68]. The GEM traits dataset is the core trait data used in this study and covered at least 70% of the basal area at the genus level for most plots (Supplementary Fig. 5). When GEM data were not available for a given species, this was obtained from the gap filled trait matrix from Aguirre-Gutiérrez et al.[4], who applied a Bayesian gap filling protocol resulting in a robust trait matrix, for most species in the studied plots, with a root mean square error of 0.16. The final trait dataset used for subsequent analysis covered above 90% of the basal area for most plots and traits.

Based on the above-mentioned traits, we calculated the functional diversity for each sampled vegetation plot and time period (T1 and T2). Plant functional trait diversity at the plot level was calculated using two metrics, (FDis) and RaoQ[27], which gave similar results (Supplementary Fig. 6). We selected FDis to continue our analysis because it can handle any number and type of traits, it is not strongly influenced by outliers and it is unaffected by species richness. Moreover, FDis has been shown to be relatively insensitive to the effects of under sampling[69]. We followed the equation from Laliberté and Legendre[27] to calculate the functional

diversity:

$$\text{FDis} = \frac{\sum a_j z_j}{\sum a_j},\qquad(1)$$

where $a_j$ reflects the abundance of species $j$ and $z_j$ is the distance of species $j$ to the weighted centroid $c$ which depicts the centroid of the $n$ species in trait space. The plant traits were weighted by the relative abundance of each of the species in the plot in terms of basal area (BA in m$^2$). Thus, FDis summarises the trait diversity and represents the mean distance in trait space of each species to the centroid of all species in a given community. All numeric traits were standardised during the FDis calculation.

**Taxonomic diversity calculation**. Plant species taxonomic diversity for each vegetation plot and time period (T1 and T2) was estimated by means of the Simpson diversity index, which considers the number of species present in a plot and their abundance[28,29]. The Simpson index was computed as:

$$\text{Simpson} = 1 - \sum Pi^2,\qquad(2)$$

where Pi denotes the proportion of individuals in the $i$th species in a community, with higher Simpson diversity index denoting higher diversity. The Simpson diversity index is a widely used and robust measure of diversity that accounts for species richness and number of individuals per species[29] and can be directly used to compare the plant communities of interest. We also calculated the Simpson diversity as Hill's numbers, i.e., when $q = 2$, and accounting for possible diversity underestimation in highly diverse plots as described in Chao et al.[70] using the iNext[71] package in R. We then compared the results to the traditional Simpson index computed above and obtained similar results (see Supplementary Fig. 1). Therefore we conducted further analysis with the traditional Simpson diversity index.

**Phylogenetic diversity calculation**. Phylogenetic diversity for each vegetation plot and time period (T1 and T2) was calculated by constructing a phylogenetic tree using the R20100701 ultrametric tree from Phylomatic[30], with branch lengths adjusted using the default ages file[72]. Based on the resulting tree we calculated the mean pairwise phylogenetic distance (MPD), mean nearest taxon distance (MNTD) and phylogenetic distance (PD)[73] to characterise the community-level phylogenetic diversity. MPD measures the mean PD matrix between communities. We used a null model based on frequency, which randomised community data abundances within species, while maintaining the same species occurrence frequency. MNTD, was calculated as the average of the smallest PD for each species to its closest relative in a given forest community. PD was calculated as the sum of the phylogenetic branch lengths of co-occurring species. The three phylogenetic diversity metrics showed the same pattern of change along the climatic gradient (milder for PD; Supplementary Fig. 6), therefore we selected only MPD for further analysis. We carried out the same analysis as above using the phylogenetic tree of Zanne et al.[74] and observed that the phylogenetic diversity values obtained for the first and second censuses were highly similar to those from the R20100701 ultrametric tree ($R^2 = 0.90$ and 0.86, respectively), thus we carried out all further analysis using the results derived from the R20100701 ultrametric tree. We tested if the above-mentioned functional traits (only for quantitative traits) show phylogenetic signal using the Blomberg's $K$ statistic[75] and assessed its significance by randomising the tree tips 999 times and comparing the resulting values to the original ones. The $K$ statistic measures the variance of a trait regarding the variance expected under a Brownian motion model with values of 0 depicting no phylogenetic signal and 1 showing strong phylogenetic signal. Phylogenetic signal analyses were carried in the R platform (v. 3.4.1)[76] using the Phylosignal package.

All diversity analyses were carried out in the R platform (v3.4.1)[76], using the 'FD'[27], 'Vegan'[77], 'Picante'[78] and 'Phytools'[79] packages.

**Climatic and soil data**. To investigate the role that climate may play on determining changes in the three facets of diversity, we gathered gridded data on potential evapotranspiration (PET in mm), precipitation accumulation (mm) and VPD from the TerraClimate project[80] at a spatial resolution of ~4 × 4 km. Using the TerraClimate data we calculated the maximum climatological water deficit (MCWD) following Malhi et al.[81], the VPD (Fig. 1) and the Standardised Precipitation and Evapotranspiration Index (SPEI)[82]. The MCWD is a metric for drought intensity and severity and is defined as the most negative value of the climatological water deficit (CWD) over a year. CWD is defined as precipitation ($P$) (mm/month) – PET (mm/month) with a minimum deficit of 0. Then:

$$\text{MCWD} = \min(\text{CWD1}\dots\text{CWD12}).\qquad(3)$$

The SPEI incorporates monthly information on temperature, precipitation and PET to calculate drought severity based on the drought intensity and duration. We calculated the SPEI based on a 12-month time window. To characterise the climatic conditions for each of the two time periods, we used a climatology of 30 years preceding each vegetation census as follows: for the first period we captured the climatic metrics during the preceding 30 years of the first census, thus between 1964 and 1993, and for the second census this time window corresponded to the years between 1984 and 2013. Based on these two time periods we also calculated

the absolute change in the MCWD, SPEI and VPD. Lastly, we calculated the MCWD, SPEI and VPD for the full term covering 1964–2013. We used a climatology of 30 years as suggested by the World Meteorological Organization (WMO) in order to characterise the average weather conditions for a given area (www.wmo.int/pages/prog/wcp/ccl/faqs.php).

Soil data was collected at the plot level between 2007 and 2013 (Supplementary Table 3). For further information on soil characteristics and sampling across the study area see Moore et al.[67] and the ForestPlots database (www.forestplots.net). We used the averaged soil characteristics (Supplementary Table 3) for the first 30 cm depth and carried a principal component analysis using the prcomp function of the stats package in R[76]. We used the first three principal component axes as they explain at least 10% of the variance, the three together explain most variance in the data (76.2%) and axis four and onwards explain <10% of data variance (Supplementary Fig. 7). The first PC was mainly loaded by cation exchange capacity, Mg and soil Nitrogen and is thus referred to as a cations-nitrogen axis; the second was mainly loaded by the soil pH, Fe and Ca and is thus referred to as an acidity-calcium axis; the third was mainly loaded by the soil texture characteristics as percentage of Clay and Sand and is thus referred to as a soil texture axis.

**Statistical analysis**. We calculated the temporal changes in functional, taxonomic and phylogenetic diversity at the plot level as the annual rate of change ($\Delta$FDis$_r$, $\Delta$Simpson$_r$ and $\Delta$MPD$_r$) as to standardise for different time between censuses for different plots. To this end we subtracted the diversity level of the first time period (T1) from that of the second time period (T2) and divided the result by the time between censuses for each vegetation plot.

To investigate if different forests communities (drier vs wetter) differ in their changes in the three facets of diversity we first we carried out a Bayesian version of a typical $T$-test analysis. We grouped the vegetation plots as belonging to the drier (MCWD in T1 $\le -250$ mm) or wetter sites (MCWD $> -250$ mm) depending on their MCWD on the recent time period. The MCWD threshold was selected as it may represent a transition from a tropical wet forest vegetation towards a more seasonal and savanna like environment as has been shown in recent studies for the Amazon[81] and West Africa[4,81] tropical forests. Then using Bayesian estimation[31,32] in a similar way than a $T$-test for a pair of observations we investigated if and to what extent the average change in each of the three facets of diversity in the drier group differed from the wetter group. We carried out the Bayesian analysis using the 'BEST' package for R[31,32], with normal priors with mean for $\mu$ of 0 and the standard deviation for $\mu$ of 10. We used broad uniform priors for $\sigma$, and a shifted-exponential prior for the normality parameter $v$.

Subsequently, we modelled the observed rate of changes in each of the three facets of diversity ($\Delta$FDis$_r$, $\Delta$Simpson$_r$ and $\Delta$MPD$_r$) as a function of the climatic variables specified above and soil characteristics (three first PCA axes). As some plots were smaller than 1 ha (Supplementary Table 1) we included plot size as a covariate in the statistical models to account for its possible effect in the observed changes in the three facets of diversity. We modelled the changes in the three facets of diversity using linear models with a Gaussian error structure under a Bayesian framework. To prevent model over parameterisation and overfitting we first calculated the Pearson's correlation coefficients between the climatic and soil variables and from each pair of those with correlation values |>0.7| we selected the most ecologically meaningful for our study and excluded the other. With this procedure we avoided distorting model coefficients in the modelling stage[83]. After correlation analysis the selected climatic and soil variables used in further analysis were the full-term MCWD (MCWD$_{\text{Full}}$), its absolute change ($\Delta$MCWD$_{\text{Abs}}$) and the absolute change in $\Delta$VPD$_{\text{Abs}}$ and the three PC soil axes (Supplementary Table 3). The statistical models were run with normal diffuse priors with mean 0 and 2.5 standard deviation for coefficients and 10 standard deviation for the intercept, three chains and 2000 iterations. We started with a model that included all environmental variables, under the hypothesis that both climate and soil play a role on the distribution of plant traits[22,84]. From this initial full model, we constructed a series of simpler models that included interactions between climatic and soil covariates (35 models in total; Supplementary Table 4). Based on leave-one-out cross-validation (LOO) we selected the model (best model) with the lowest LOOIC (LOO Information Criterion) score and highest expected log predictive density difference[85]. We computed the highest density intervals (HDI) rendering the range containing the 89% most probable effect values as suggested in Makowski et al.[86] and calculated the ROPE values using such HDI. Although the 95% HDI was not used as this range has been shown to be unstable with ESS < 10,000 (effective sample size)[32] we also present it together with the 50% HDI as to give a more complete description of the data. We calculated the region of practical equivalence (ROPE)[87] to test the importance of parameters, where if the ROPE is 0 or close to 0 it gives strong indication of the important effect that a given explanatory variable has on the response variable. In the results section we discuss the results based on the first best model obtained and give details of all models in the Supplementary information (Supplementary Table 6). All environmental variables were scaled and centred prior to model fitting. We conducted all statistical analysis in R (v. 3.4.1)[76] using the, 'BEST'[88], 'rstanarm'[89], 'loo'[90] and 'bayestestR'[86] packages.

**Reporting summary**. Further information on research design is available in the Nature Research Reporting Summary linked to this article.

## Data availability

The vegetation census and plant functional traits data that support the findings of this study are available from their sources (www.ForestPlots.net and gem.tropicalforests.ox.ac.uk/). The processed community-level data used in this study is available in the following repository: https://doi.org/10.6084/m9.figshare.12251378.

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

## Acknowledgements

This work is a product of the Global Ecosystems Monitoring (GEM) network (gem.tropicalforests.ox.ac.uk), the African Tropical Rainforest Observation Network (Afri-TRON; www.afritron.org) and ForestPlots.net. J.A.G. was funded by the Netherlands Organisation for Scientific Research (NWO) under the Rubicon programme with project number 019.162LW.010. The trait field campaign was funded by a grant to Y.M. from the European Research Council (Advanced Grant GEM-TRAIT: 321131) under the European Union's Seventh Framework Programme (FP7/2007-2013), and from the Royal Society-Leverhulme Africa Capacity Building Programme. The long-term forest monitoring campaigns were funded by Natural Environment Research Council (NERC) grants NE/I014705/1 and NE/P001092/1 to Y.M., a Royal Society Research Fellowship to S.L.L., University of Leeds scholarship to S.F., and a European Research Council (Advanced Grant T-FORCES:291585) to O.L.P., Y.M. and S.L.L. ForestPlots.net data management were funded by grants to S.L.L. from the Royal Society, the University of Leeds PhD studentship to S.F., from the European Research Council (Advanced Grant T-FORCES: 291585) to O.L.P., Y.M. and S.L.L. During data collection, I.O. was supported by a Marie Curie Fellowship (FP7-PEOPLE-2012-IEF-327990), and S.F. was supported by a University of Leeds Earth and Biosphere institute studentship. Plot inventory data were funded by the Royal Society, NERC, Sebright's Education Foundation and Gilchrist Educational Trust. The authors thank Michael D. Swaine for his contribution with vegetation plot data and Natascha Luijken for her assistance with vessel measurements. Y.M. is supported by the Jackson Foundation.

## Author contributions

J.A.G. conceived the idea of the study and together with I.O. conceived the phylogenetic analysis framework. Y.M. conceived and funded the traits field campaign and associated field plots, I.O., A.G., T.P. collected trait data, I.O., K.Z., A.G. processed laboratory samples and cleaned and parsed the GEM trait data. S.L.L. conceived the AfriTRON long-term plot network measurement recensus programme. S.L.L., S.M., S.F., W.H., S.A.B., T.P. and K.A.B. collected the plant census data, and S.L.L., O.L.P. and Y.M. funded their collection. J.A.G. designed and carried out the analysis and wrote the first version of the paper. I.O. carried out the initial phylogenetic analysis and together with Y.M. discussed and commented on the first draft of the paper. All authors discussed the results and commented on later versions of the paper and approved the final version.

## Competing interests

The authors declare no competing interests.
