## [Peer Review File · Nature Communications]

Reviewers' comments first round:

Reviewer #1 (Remarks to the Author):

In this manuscript Aguirre-Gutiérrez et al expand on their recent analysis of temporal shift in the functional structure of tree communities in Western Africa (<https://onlinelibrary.wiley.com/doi/full/10.1111/ele.13243>) by exploring temporal changes in different diversity facets. They found that the impact of drier conditions on tropical tree diversity were depending on historical conditions: wet forests reached higher levels of taxonomic and functional diversity while dry forest showed decline in taxonomic and functional diversity. These results are important to derive better scenarios of changing diversity following climate change in the undersampled areas of tropical forests.

Following the editor request I focused on the data analysis and have a couple of major comments:

- the separation between wet and dry forest using a cut-off of -250mm MCWDFull is not justified in the manuscript and eyeballing the data does not seem to give support to this artificial split. Looking at the figure 1 it seems that rather than being two separate groups in the data (wet and dry), the sampled plots appear to form a continuous gradient in water deficit. The authors should justify the use of the -250mm cut-off, failing that the first set of analysis (using the Bayesian t-test) should be re-ran this time with a continuous water deficit (but see next comment).

- the first analysis will require more description in the method section. Specifically: name the R-package used, describe the essence of the analysis (saying „we use Bayesian estimation“ is not informative at all), provide numerical values for the parameter of the prior distribution used (ie we used a normal prior with mean 0 and standard deviation 2). This analysis use a Bayesian t-test to compare the changes between the wet and dry plots, I might have missed it but the time between the surveys was not standardized and could potentially introduce a bias in the analysis (if wet plots have longer time between re-census than dry plots). I would recommend to drop this analysis and just use the linear models developed in the second part. From these the predicted changes for all diversity metrics could be derived while taking into account potential bias (such as differing soil conditions and different time between census).

- model selection is a tricky endeavour, some design variables (such as time between census) might be insignificant and might not improve model fit, but one should still leave these in to control for their effect. Rather than doing a blind model selection of all potential covariates, I would recommend to keep a core set of design variables that should be present in all models and that should be used to control for their impact on the diversity changes.

So this manuscript has some potential but the analysis should be streamlined to better fit the question and the data at hand.

Minor comments:

line 69-70: Sentence unclear, re-phrase

line 75: Not familiar with the concept of species replaceability, but does this means that some species are expendable and that their extinciton should not be a cause of concern?

Line 87-88: The current wording make it sounds like that forests are biological entities capable of adaptation, some re-wording is needed.

Line 93: remove „facets“

Line 112-136: I am not familiar with the Nature manuscript format (no method sciton in main

text), but this paragraph seems pretty detailed and methody for an introduction. Part of it could be removed to improve the textual flow.

Line 123: Word out MCWD

Line 144: Give credible intervals around that estimate, also throughout this section give the estimate changes (and credible intervals) for both dry and wet forests

Line 150: It is confusing to mention an average that is larger and to report a negative coefficient estimate.

Line 152: And what about overall changes in taxonomic and functional diversity?

Line 153-155: Confusing sentence, re-write.

Line 176-178: Give some estimates of functional diversity decrease in the areas with strongest and smallest drying out

Line 182-183: Give some estimate of the increase in phylogenetic diversity with soil PC1

Line 198: Please word out the new fundamental knowledge created by your manuscript

Line 267-269: from the results it is hard to find support for an adaptation of species to drier conditions, for this you could show how some trait values shifted towards more drought-tolerant values (like lower LA:SA). Simple bivariate graphs of changes in these traits versus MCWD variations would be very informative.

Line 333-336: Were species measured at multiple sites along the precipitation gradient? And if yes was this taken into account when computing the Fdis? Like using trait values from dry conditions when computing Fdis in a dry plot

Line 397-398: Provide appropriate reference to the R packages used

Line 441: Why make the break at -250mm? Is there some theoretical expectations? Is the distribution of the MCWD warranting such a break (ie is it bimodal with a low around -250), would be nice to see an histogram of the distribution of MCWD at T1 to justify this break

Line 443-447: More information is needed on the modelling approach used here: Which model was fitted? With which software? Which prior were used (saying broad uniform prior is not enough, specify the bounds of the uniform prior)? And also if you derived HDI why not use them when presenting the results?

Line 448: explicitly state how „change“ was computed (like recent – ancient)

Line 461: Which deviation / precision had these priors? Where they applied to all model parameters?

Line 466: How far in terms of LOOIC was the best model from the second to best model? From Table S6 it seems that for the Fdis, model 7-14-3 all have LOOIC values that are overlapping when subtracting their standard error. In that regard focusing on the variables present only in the best model (nb 7) seems a bit arbitrary

Table 1: Explain the ROPE indicator and how it was computed in the method section

Figure 1: It would be best to use the same variable on the y-axis of the left plots as the one used in the models, so plotting the change against the climate. Also it would be most interesting if these graphs showed the relation presented in Table 1, like change in Fdis ~ variation in precipitation (actual observations as dots and fitted regression line with credible interval or HDPI). It is interesting to note that the drier sites (to the left of the figure), tended to experience the strongest

drying, maybe a SI figure could plot these two variables together. In panels d-f please also provide HDPI / credible intervals for the average diversity levels in T1 and T2. It is also confusing that the insets report the comparison of the changes between the two period which is not immediately obvious. Another way to present this would be to have a dot with HDPI for the estimated changes in wet and dry forest and then a probability reporting whether changes in wet forests are bigger than changes in dry forests.

Figure 2: Figure 1 a-c could be replaced by this plot. The color filling is redundant in panels a and b.

Figure 3: Some plots appear to be very close to each other, have you checked for spatial autocorrelation in the model residuals?

Lionel Hertzog

Reviewer #2 (Remarks to the Author):

Comments on MS NCOMMS-19-26656377 entitled "Long-term droughts may drive drier tropical forests towards increased functional, taxonomic and phylogenetic homogeneity" by Aguirre-Gutiérrez et al.

I carefully read the paper and all comments and suggestions from the first and second round reviewers. I agree that this paper is novel in explaining the variation of three facets of tropical forests, and its importance for prediction and management of forest diversity change. I only have two major concerns:

1) The biodiversity change was calculated based on the 21 1-ha forest plots along a rainfall gradient from 2000 mm to 700 m. I guess there is also a gradient of species richness along this rainfall gradient. The biodiversity change in these forests not only depends species richness and evenness but also sampling completeness (Chao, A., & Jost, L. 2012). This gradient may bias toward underestimation of biodiversity in highly diverse forests. This may lead to bias in estimation of biodiversity change. Maybe it is possible to correct the bias using their package iNEXT. Further, Simpson index is a special case of Hill number when $q=1$ which gives more weight for abundant species. So this means that the biodiversity change mainly referred to variation of abundant species. For phylogenetic diversity, authors need clearly state they used null model to standardize phylogenetic diversity.

2) Authors expect "soils with higher water holding capacity may buffer drought impacts on forest communities". "forest soils high in clay may be able to maintain higher water availability over longer periods during droughts than sandy soils where the water holding capacity tends to be lower" (Line 61-62). "PC3 was mainly loaded by the soil texture characteristics as percentage of Clay and Sand and is thus referred to as a soil texture axis." (Lines 429-430). However, they found that PC1 (mainly representing cation exchange capacity, exchangeable magnesium (Mgex) and nitrogen content) were correlated with phylogenetic diversity change. I am not familiar with how soil water holding capacity relates to soil nutrients. I guess there are some missing link here.

References:

Chao, A., & Jost, L. (2012). Coverage-based rarefaction and extrapolation: standardizing samples by completeness rather than size. *Ecology*, 93(12), 2533-2547.

Response to comments from reviewers
Nature Communications

Long-term droughts may drive drier tropical forests towards increased functional, taxonomic and phylogenetic homogeneity

Jesús Aguirre-Gutiérrez, Yadvinder Malhi, Stephen Adu-Bredu, Kofi Affum-Baffoe, Timothy R. Baker, Sophie Fauset, Agne Gvozdevaite, Wannes Hubau, Simon L. Lewis, Sam Moore, Theresa Peprah, Kasia Ziemińska, Oliver L. Phillips, Imma Oliveras

Response to Comments from Referee 1

General comment:

‘...These results are important to derive better scenarios of changing diversity following climate change in the undersampled areas of tropical forests.’

Response:

Dear Dr. Lionel Hertzog. Thank you for your detailed review of our manuscript and for considering our work important for disentangling biodiversity responses to changing climate, specially in undersampled regions such as tropical forests.

R1-1: The separation between wet and dry forest using a cut-off of -250mm MCWDFull is not justified in the manuscript and eyeballing the data does not seem to give support to this artificial split. Looking at the figure 1 it seems that rather than being two separate groups in the data (wet and dry), the sampled plots appear to form a continuous gradient in water deficit. The authors should justify the use of the -250mm cut-off, failing that the first set of analysis (using the Bayesian t-test) should be re-ran this time with a continuous water deficit (but see next comment).

Response:

We based the selection of the threshold on our a priori knowledge of the Ghanaian forests and where at which location and climatic deficit such vegetation begins to shift from wet tropical forests towards drier vegetation and transitioning to savanna vegetation. Our threshold may thus represent a transition from a tropical wet forest vegetation towards a more seasonal and savannah like environment. Such threshold is also based on the finding from other recent studies for the Amazon (Malhi et al. 2009 -PNAS) and West Africa tropical forests (Aguirre-Gutierrez et al. 2019 -Ecology Letters).

We acknowledge that such hard threshold may not be representative of the subtle vegetation transitions one may find in nature and this is the reason why we use this first analysis to contrast possible shifts in diversity these finite groups may show across long periods of time after a long term drought. In a second stage we model such changes in diversity along the climatic gradient, thus without grouping the forests, as a response to a changing climate. Therefore, we show both responses, as finite groups (wet vs dry forests) and also as a vegetation continuum (plots along the climatic gradient without grouping), which strengthens our results and conclusions as both results point towards the stronger decline in diversity in drier forests than in wetter forests.

R1-2: The first analysis will require more description in the method section. Specifically: name the R-package used, describe the essence of the analysis (saying „we use Bayesian estimation“ is not informative at all), provide numerical values for the parameter of the prior distribution used (ie we used a normal prior with mean 0 and standard deviation 2). This analysis use a Bayesian t-test to compare the changes between the wet and dry plots, I might have missed it but the time between

the surveys was not standardized and could potentially introduce a bias in the analysis (if wet plots have longer time between re-census than dry plots).

Response:

We have including the information requested in the methods section from the manuscript, including full references to the cited work as suggested, see below for the new text.

'First we calculated the temporal changes in functional, taxonomic and phylogenetic diversity at the plot level as the annual rate of change ($\Delta FDis_r$, $\Delta Simpson_r$ and ΔMPD_r) as to standardise for different time between censuses for different plots and carry at a Bayesian version of a typical T-test analysis. To this end we subtracted the diversity level of T1 from that of T2 and divided the result by the time between censuses for each vegetation plot. Then we grouped the vegetation plots as belonging to the drier (MCWD in T1 \leq -250mm) or wetter sites (MCWD > -250mm) depending on their maximum climatic water deficit (MCWD) on the recent time period. The MCWD threshold was selected as it may represent a transition from a tropical wet forest vegetation towards a more seasonal and savannah like environment as has been shown in recent studies for the Amazon⁷⁴ and West Africa⁴ tropical forests. Then using Bayesian estimation^{30, 31} in a similar way than a T-test for a pair of observations we investigated if and to what extent the average change in each of the three diversity facets in the drier group differed from that of the wetter group. We carried out the Bayesian estimation using the 'BEST' package for R using the default values for diffuse priors as suggested by Kruschke^{30, 31}, with normal priors with mean of the prior of μ set to the mean of the pooled data and the standard deviation for μ to 1,000 times the standard deviation of the pooled data as to keep the prior scaled appropriately relative to the scale of the data. We used broad uniform priors for σ , and a shifted-exponential prior for the normality parameter v . We computed the highest density intervals (HDI) rendering the range containing the 89% most probable effect values as suggested by Kruschke³¹ as this results in more stable predictions.'

About the comment 'it but the time between the surveys was not standardized and could potentially introduce a bias in the analysis':

Following the comment from the reviewer we reanalysed the diversity change data for the Bayesian version of the T-Test with the 'BEST' R package and present this new analysis in our manuscript instead of the old one. Although the difference in time between censuses for the different plots is not large and there was no relationship between the changes in the diversity facets and such metrics we agree that it is important to account for it in this analysis. In this new version of our analysis we compute the rate of change in the three diversity facets and investigate if the drier and wetter plots differ in such rate of change, which accounts for the time between censuses (see below for new methods). Our results did not change in essence, as you can see in our new Figure 2 and Supplementary Figure 1.

R1-3 And related comment

Figure 1: It would be best to use the same variable on the y-axis of the left plots as the one used in the models, so plotting the change against the climate. Also it would be most interesting if these graphs showed the relation presented in Table 1, like change in Fdis \sim variation in precipitation (eactual observations as dots and fitted regression line with credible interval or HDPI). It is interesting to note that the drier sites (to the left of the figure), tended to experience the strongest drying, maybe a SI figure could plot these two variables together. In panels d-f please also provide HDPI / credible intervals for the average diversity levels in T1 and T2. It is also confusing that the insets report the comparison of the changes between the two period which is not immediately

obvious. Another way to present this would be to have a dot with HDPI for the estimated changes in wet and dry forest and then a probability reporting whether changes in wet forests are bigger than changes in dry forests. I would recommend to drop this analysis and just use the linear models developed in the second part. From these the predicted changes for all diversity metrics could be derived while taking into account potential bias (such as differing soil conditions and different time between census).

Response:

Following the comment from the reviewer we improved our figure showing the change in the three diversity facets, now Figure 2. We followed the advise of the reviewer and now we show in Figure 2d-f the average change in wet and dry forests as a point with the HDI for each one of the three diversity facets. We also include the inset in each panel (d-e-f) stating the average difference in rate of change values between drier and wetter forests together with probability and HDI values for each diversity facet as suggested by the reviewer. Now in our new Figure 3 we continue plotting the observed change in the three diversity facets against the absolute change in the maximum climatic water deficit for the functional and taxonomic diversity metrics.

We thus believe that our analysis contrasting wetter and drier forests are of importance as to show how both groups may behave in a contrasting manner after long term droughts. Moreover, such analysis help answering our first research questions: 1) *if and to what extent there have been shifts in the three diversity facets across time*. Such findings are also corroborated by our second set of analysis, which respond to our questions, *'to what extent the shifts in the three diversity facets are explained by changes in climate?'* showing the general forests responses to a changing climate without dividing the forest by their MCWD.

R1-4 Model selection is a tricky endeavour, some design variables (such as time between census) might be insignificant and might not improve model fit, but one should still leave these in to control for their effect. Rather than doing a blind model selection of all potential covariates, I would recommend to keep a core set of design variables that should be present in all models and that should be used to control for their impact on the diversity changes.

Response:

We agree that model selection should not be done blindly but should be done with essential ecological background and hypothesis. We state in our manuscript our expectations and hypothesis: *'We expect that a drying trend would be reflected in overall diversity decreases along the water deficit gradient, however, forest communities located in the drier end of the water deficit gradient may experience higher climatic stress and therefore the diversity changes may be stronger in those locations. Responses in the three diversity facets may be determined by soil characteristics in addition to climatic conditions; for example, soils with higher water holding capacity may buffer drought impacts on forest communities^{7,25}. Therefore, we also investigate the role of soil characteristics on the response of the three diversity facets along the climatic gradient and across time.'*

Based on such expectations we built the set of models outlined in Supplementary Table 4.

In our statistical models we try to avoid overparameterization and decreasing degrees of freedom by excluding variables that we know a priori may not have an effect in the response variable (e.g. 'time between censuses'; Supplementary Figure 2). However, we agree with the reviewer that it may be worth including it in the model as to be sure it does not affect the overall output. Therefore, following his advice we included the covariate 'time between censuses' in our models of Taxonomic diversity as a test and we did not find changes in the outcome, with the same models being the best one at the top of the list (see Table 1 below). We also ran the best and null model for the FDis response variable (Table 2 below) and found the same pattern (see Figure 1 below). Therefore, we are now sure the time between censuses has no effect in our models and including it would only

decrease degrees of freedom with added value. Thus, we avoided including this covariate in the models shown in the manuscript.

Table 1. Results for models of Simpson diversity change as a function of climate and soil conditions including time between census as a covariate

	elpd_diff	se_diff	elpd_loo	se_elpd_loo	p_loo	se_p_loo	loaic	se_loaic
si_7	0.0000	0.0000	73.6218	3.2139	3.5925	0.9327	-147.2437	6.4278
si_5	-0.6450	2.3883	72.9769	2.7974	3.1561	0.7437	-145.9538	5.5948
si_23	-1.1578	2.1560	72.4640	3.3563	2.5233	0.7669	-144.9281	6.7127
si_6	-1.1721	1.7686	72.4497	3.3557	3.2602	0.9432	-144.8995	6.7114
si_3	-2.0640	2.0271	71.5578	3.5444	6.1570	1.8719	-143.1157	7.0888
si_9	-2.2638	1.7949	71.3581	2.6500	3.7129	0.9790	-142.7161	5.3000
si_15	-2.6022	1.6096	71.0196	2.8613	5.8516	1.1977	-142.0392	5.7225
si_8	-2.9535	2.2794	70.6683	2.9657	4.4965	1.3350	-141.3367	5.9314
si_17	-3.1751	2.6380	70.4468	2.4095	6.6119	1.4359	-140.8935	4.8189
si_10	-3.2221	2.8840	70.3997	3.5119	5.0628	1.9739	-140.7994	7.0239
si_19	-3.2673	3.6612	70.3545	2.9556	7.6017	2.3053	-140.7090	5.9112
si_21	-3.2708	1.6133	70.3511	2.7979	5.8743	1.0630	-140.7021	5.5958
si_22	-4.0732	4.5626	69.5486	4.0101	9.2839	3.1927	-139.0972	8.0203
si_24	-4.2648	2.2145	69.3570	2.3419	5.4964	1.2106	-138.7140	4.6838
si_18	-4.4056	2.3127	69.2162	2.2552	5.5852	1.0553	-138.4324	4.5104
si_16	-4.5949	2.7882	69.0269	3.9757	8.0656	2.4923	-138.0539	7.9513
si_14	-4.9664	2.5519	68.6554	3.8432	8.2687	2.4281	-137.3109	7.6863
si_25	-5.4182	2.9597	68.2036	3.3007	6.9470	2.1263	-136.4072	6.6013
si_26	-5.5112	2.7934	68.1106	3.3829	7.1362	2.3207	-136.2212	6.7659
si_20	-5.7653	2.3804	67.8565	3.3094	7.5244	2.0519	-135.7130	6.6188
si_30	-6.4058	3.0665	67.2160	3.6344	10.3808	2.4834	-134.4321	7.2687
si_35	-6.4300	2.9470	67.1918	2.4402	10.4031	1.9251	-134.3837	4.8804
si_34	-7.6579	4.2374	65.9639	3.0962	12.5061	2.5573	-131.9278	6.1923
si_29	-7.7585	3.9194	65.8634	3.1428	12.7995	2.6179	-131.7267	6.2855
si_4	-8.1146	2.9647	65.5073	3.2032	8.9888	2.3742	-131.0145	6.4064
si_33	-8.5797	3.7875	65.0422	2.6454	10.9223	1.9741	-130.0843	5.2907
si_27	-9.2067	1.8849	64.4151	3.0994	10.4211	2.0323	-128.8303	6.1989
si_28	-9.3016	4.3704	64.3202	3.6418	14.4263	3.2710	-128.6404	7.2836
si_1	-10.7091	3.4960	62.9127	3.7603	12.7370	2.9674	-125.8254	7.5205
si_32	-11.0591	3.3407	62.5628	4.3938	13.0336	3.5065	-125.1255	8.7875
si_31	-11.3641	3.7917	62.2578	4.6960	13.6008	3.5167	-124.5155	9.3920
si_11	-13.1253	3.5837	60.4966	2.8251	17.9338	2.6158	-120.9931	5.6501
si_2	-15.4123	3.8740	58.2096	2.8395	49.5649	2.8359	-116.4191	5.6791
si_13	-15.9349	4.3009	57.6870	4.5819	17.6635	4.0479	-115.3739	9.1638
si_12	-17.4949	4.9155	56.1269	4.2296	21.0488	3.9203	-112.2538	8.4591

Table 2. Results for best models and random model of Functional diversity change as

a function of climate and soil conditions including time between census

	elpd_diff	se_diff	elpd_loo	se_elpd_loo	p_loo	se_p_loo	looic	se_looic
FDis_7	0.0000	0.0000	83.4005	2.8308	3.6050	1.1619	-166.8011	5.6616
FDis_14	-1.3973	1.1590	82.0032	2.9618	5.6630	1.4268	-164.0065	5.9235
FDis_3	-2.2126	0.8981	81.1879	2.8459	5.3707	1.4705	-162.3758	5.6919
FDis_23	-3.2794	2.3139	80.1211	2.8778	2.3996	0.8181	-160.2422	5.7556
FDis_16	-3.3111	1.5738	80.0895	2.3971	6.0889	1.2141	-160.1789	4.7942

Figure 1. Best model results for change in Simpson diversity (first panel) and Functional diversity (second panel) as a function of Maximum Climatic Water Deficit (MCWD) and time between censuses. Here, it is evident the lack of effect of this variable, which is the pattern found for the rest of the models built.

R1 Minor comments:

R1-5

line 69-70: Sentence unclear, re-phrase

Response:

Following the suggestion from the reviewer we have rephrased the sentence as follows:

'It has become evident the role that high functional and phylogenetic diversity levels may play for increasing the ecosystems resilience to changes in environmental conditions.'

R1-6

line 75: Not familiar with the concept of species replaceability, but does this means that some species are expendable and that their extinciton should not be a cause of concern?

Response:

In this context it refers to how alike or how related (phylogenetically) is the given species to the others present in the community and not necessarily to the extinction of the species in a given region or globally.

R1-7

Line 87-88: The current wording make it sounds like that forests are biological entities capable of adaptation, some re-wording is needed.

Response:

The sentence has been rephrased as follows:

'In such drier forests the abundance of deciduous species is increasing, which could be generating forest communities better adapted to a drying climate⁵.'

R1-8

Line 93: remove „facets“

Response:

The word 'facets' has been removed.

R1-9

Line 112-136: I am not familiar with the Nature manuscript format (no method section in main text), but this paragraph seems pretty detailed and methody for an introduction. Part of it could be removed to improve the textual flow.

Response:

We have reduced this sections and now it is as follows:

'We analyse the functional, taxonomic and phylogenetic diversity of 21 unique vegetation plots (Fig. 1) across two time periods (range 1987-2013; Supplementary Table 1) and calculate their changes in functional diversity (FDis)²⁶ using a comprehensive dataset of more than 1500 single plant trait samples from 18 functional traits (Supplementary Table 2). Taxonomic diversity was estimated by means of the Simpson diversity index^{27, 28}, and phylogenetic diversity was calculated as the mean pairwise phylogenetic distance between all individuals in the community (MPD)²⁹. To assess shifts in the three diversity facets across time and along the climatic gradient we used Bayesian estimation^{30, 31}. To investigate the role that climate may play on determining changes in the three diversity facets we calculated the mean maximum climatic water deficit (MCWD) and vapour pressure deficit (VPD) for the full term of the study (MCWD_{Full} and VPD_{Full} respectively) and also characterised the water availability for each census time. We then calculated the absolute and relative changes for each metric between the first and second periods (Δ MCWD_{Abs}, Δ MCWD_{Rel} and Δ VPD_{Abs}, Δ VPD_{Rel}). We conducted a principal component analysis of the soil information and extracted the first three PCA axes for further analysis (Supplementary Table 3). To test for the effects of climatic and soil conditions on the changes in the three diversity facets we constructed different statistical models under a Bayesian framework (Supplementary Table 4) and selected the most parsimonious models based on their leave one out information criterion (LOOIC) and expected log predicted density (ELPD).'

R1-10

Line 123: Word out MCWD

Response:

This has been now written in full.

R1-11

Line 144: Give credible intervals around that estimate, also throughout this section give the estimate changes (and credible intervals) for both dry and wet forests

Response:

As to not repeat information and improve textual flow, instead of writing all mean and credible we now state the number of the table and/or figure where the details of mean and credible intervals of all models are specified.

R1-12

Line 150: It is confusing to mention an average that is larger and to report a negative coefficient estimate.

Response:

We have updated this paragraph based on the new results carrying out the analysis suggested by the reviewer. Our new text focusing on the comment from the reviewer is as follows:

'The phylogenetic diversity (MPD) showed large average decreases along the climatic gradient (Fig. 2c), with forests at the drier end of the water deficit spectrum showing on average larger ($\mu=-0.03$) but not statistically different rates of phylogenetic diversity declines than forests in wetter locations (Probability= 62.2%; Fig. 2f).'

R1-13

Line 152: And what about overall changes in taxonomic and functional diversity?

Response:

This is now specified as:

'In summary, the drier forests are transitioning towards increasingly more homogenous forest communities, diverging further from wetter forests in functional, taxonomic and phylogenetic diversity.'

R1-14

Line 153-155: Confusing sentence, re-write.

Response:

This has been rewritten as:

The phylogenetic and functional diversity changes were not significantly correlated (Supplementary Fig. 2) even though all traits that conform the functional diversity metric (FDIs), showed significant phylogenetic signal (Supplementary Table 5).

R1-15

Line 176-178: Give some estimates of functional diversity decrease in the areas with strongest and smallest drying out

Response:

We now include the suggested details:

'Functional diversity decreased the most (up to -0.0072 for plot BBR_16) in areas that experienced the strongest negative Δ MCWDAbs (-27.5 mm; Fig. 3a) and increased (up to 0.009 for plot KDE_2) in areas that experienced the smallest Δ MCWDAbs (-7.5 mm; $R^2_{adj}=0.36$; Table 1).'

R1-16

Line 182-183: Give some estimate of the increase in phylogenetic diversity with soil PC1

Response:

We now include the suggested details:

'Forest communities in areas where soils were low in eCEC, Mg_{ex} and nitrogen (6 out of 21) slightly increased their phylogenetic diversity (MPD up to 2.3 for plot BOR_06), however, most forest communities (15 vegetation plots) decreased in phylogenetic diversity (up to -8.2 for plot BBR_17), with the strongest decrease shown for communities richer in soil eCEC, Mg_{ex} and soil nitrogen ($R^2_{adj}=0.53$; Fig. 3c; Table 1).'

R1-17

Line 198: Please word out the new fundamental knowledge created by your manuscript

Response:

We have improved the text as follows:

'The observed shifts in diversity facets across tropical forest communities provide a fundamental advance in our understanding of how forests may respond under a drying climate, showing that such responses may depend on the forest communities position along the climatic gradient and the changes in water availability experienced across time.'

R1-18

Line 267-269: from the results it is hard to find support for an adaptation of species to drier conditions, for this you could show how some trait values shifted towards more drought-tolerant values (like lower LA:SA). Simple bivariate graphs of changes in these traits versus MCWD variations would be very informative.

Response:

We followed the advice of the reviewer and updated the information relating shifts in single functional traits to climate we have already described in Aguirre-Gutierrez et al. 2019 Ecology Letters. Now, we directly mention the changes in single traits that we have found for drier tropical forest in the same study area and have updated the text as follows:

'Our results suggest that drier forest communities are changing their functional trait composition in part as a response to a drying climate. Such changes are selecting for species better adapted to drier conditions as shown by the already observed increases in the abundance of deciduous species, with lower leaf area : sapwood area ratios and higher photosynthetic capacity in West African tropical forests⁵.'

R1-19

Line 333-336: Were species measured at multiple sites along the precipitation gradient? And if yes was this taken into account when computing the Fdis? Like using trait values from dry conditions when computing Fdis in a dry plot

Response:

The traits collection was carried out at seven different 1 ha plots across the climatic gradient. This trait data covers at least 70% of the basal area at the genus level for most plots. Therefore, some species that are abundant in a given plot and were sampled for traits may not be abundant in another plots (i.e. account for a very small portion of the basal area from a plot) and may not be sampled in such plot. Therefore, our approach is to get an average trait value per species based on all samples obtained for that given species.

R1-20

Line 397-398: Provide appropriate reference to the R packages used

Response:

References have been provided for R packages used.

R1-20

Line 441: Why make the break at -250mm? Is there some theoretical expectations? Is the distribution of the MCWD warranting such a break (ie is it bimodal with a low around -250), would be nice to see an histogram of the distribution of MCWD at T1 to justify this break

Response:

Please see full response to the main comment (R1-3) on this topic made above.

R1-21

Line 443-447: More information is needed on the modelling approach used here: Which model was fitted? With which software? Which prior were used (saying broad uniform prior is not enough, specify the bounds of the uniform prior)? And also if you derived HDI why not use them when presenting the results?

Response:

Please see full response to the main comment (R1-2) on this topic made above.

R1-22

Line 448: explicitly state how „change“ was computed (like recent – ancient)

Response:

We now explicitly state in the text how the change was computed.

R1-23

Line 461: Which deviation / precision had these priors? Where they applied to all model parameters?

Response:

We used diffuse priors, so weakly informative. Following the comment from the reviewer we have updated our manuscript as to improve the description of the parameters of the models as follows: *‘The statistical models were run with normal diffuse priors with mean 0 and 2.5 standard deviation for coefficients and 10 standard deviation for the intercept, three chains and 2000 iterations. We started with a model that included all environmental variables, under the hypothesis that both climate and soil play a role on the distribution of plant traits’.*

R1-24

Line 466: How far in terms of LOOIC was the best model from the second to best model? From Table S6 it seems that for the Fdis, model 7-14-3 all have LOOIC values that are overlapping when subtracting their standard error. In that regard focusing on the variables present only in the best model (nb 7) seems a bit arbitrary

Response:

We focus on the higher performing models based on their LOOIC and LOOELPD. Following this criteria, the difference in LOOIC between the first best model and the second model for FDis is 4.1, 1.8 for Simpson diversity and 3.2 for MPD. As the difference in LOOIC for FDis and MPD is well above

two units in LOOIC we decided to focus only on the first best performing model and give details of all others in the Supplementary Table S6. The first and second best models of Simpson diversity change are less clearly differentiable based on their LOOIC (1.8 difference) but however are still different and as to be consistent with the presentation of only the first best model in the main manuscript we also selected only the first best model for the Simpson diversity index. Moreover, the contribution of the absolute change in VPD to the explanatory power of the model is low, making its effect minimally important.

Following the comment from the reviewer we now include in the main text the next text that refers to the full results of the second best models for all diversity facets:

'Results for the second best models for the three diversity facets following the leave one out cross-validation are also shown in Supplementary Table S7.'

R1-25

Table 1: Explain the ROPE indicator and how it was computed in the method section

Response:

This is now described in the text and useful references are given.

R1-26

Figure 2: Figure 1 a-c could be replaced by this plot. The color filling is redundant in panels a and b.

Response:

Please see replies to comments to R1-3. We just kept the colouring because for some readers the colours are more intuitive and help them understand the patterns of distributions of a given set of values.

R1-27

Figure 3: Some plots appear to be very close to each other, have you checked for spatial autocorrelation in the model residuals?

Response:

Yes, and we have also addressed this in a recent publication, Aguirre-Gutierrez et al. Ecology Letters 2019 22, 855-865, and we did not find a spatial autocorrelation effect between plots with a similar spatial distribution. This can be due to the fact that although some plots are closer together than others overall all plots are well distributed across the study area.

Response to Comments from Referee 2

General comment:

'I carefully read the paper and all comments and suggestions from the first and second round reviewers. I agree that this paper is novel in explaining the variation of three facets of tropical forests, and its importance for prediction and management of forest diversity change.'

Response:

Dear reviewer 2, thank you for taking the time to review not only our improved manuscript but also for reading all past reviews with such care and for agreeing that we present an important contribution for predicting and managing diversity change in tropical forests.

R2-1: The biodiversity change was calculated based on the 21 1-ha forest plots along a rainfall gradient from 2000 mm to 700 m. I guess there is also a gradient of species richness along this rainfall gradient.

Response:

At the start we also thought there would be a strong gradient of species richness along the climatic gradient. However, we saw that differences may be more marked by the identity of the species and their abundance in different plots (along the gradient) than by the species richness per se (see Fig. 2 below).

Figure 2. Species richness along the climatic gradient.

R2-2: The biodiversity change in these forests not only depends species richness and evenness but also sampling completeness (Chao, A., & Jost, L. 2012). This gradient may bias toward underestimation of biodiversity in highly diverse forests. This may lead to bias in estimation of biodiversity change. Maybe it is possible to correct the bias using their package iNEXT. Further, Simpson index is a special case of Hill number when $q=1$ which gives more weight for abundant species. So this means that the biodiversity change mainly referred to variation of abundant species. For phylogenetic diversity, authors need clearly state they used null model to standardize phylogenetic diversity.

Response:

We thank the reviewer for highlighting this point as we had not considered it. We carried out new analysis and updated our manuscript and figures as needed, see below more details. Although we sampled the full 1 ha plots for all individuals >10cm DBH and the locations of our plots are well distributed in the study area we wanted to make sure there is no effect of sampling intensity or completeness as suggested by the reviewer it may be the case. We therefore ran new analysis with the iNEXT package as suggested by the reviewer and calculated the corrected Simpson index suggested by Chao, A. and colleagues as Hill's numbers correcting for the sampling completeness. We used the function ChaoSimpson() in the iNEXT R package (Hsieh, T., Ma, K. & Chao, A. iNEXT: an R package for rarefaction and extrapolation of species diversity (Hill numbers). *Methods in Ecology and Evolution* 7, 1451-1456 (2016). We show the common Simpson index and the iNEXT results below in Table 3, where it is evident both are highly correlated. We then calculated the change in the iNext Simpson diversity using the methods described in the manuscript and found the results we obtained with the common Simpson index and the iNEXT Simpson were highly similar (see Figure 3 below).

We now also include the results of the iNEXT analysis and Figure 3 in our manuscript and have updated our text as follows:

'Plant species taxonomic diversity for each vegetation plot and time period (T1 and T2) was estimated by means of the Simpson diversity index, which considers the number of species present in a plot and their abundance^{27,28}. The Simpson index was computed as: $Simpson = 1 - \sum Pi^2$, where Pi denotes the proportion of individuals in the i th species in a community, with higher Simpson diversity index denoting higher diversity. The Simpson diversity index is a widely used and robust measure of diversity that accounts for species richness and number of individuals per species²⁸ and can be directly used to compare the plant communities of interest. We also calculated the Simpson diversity as Hill's numbers, i.e. when $q=2$, and accounting for possible diversity underestimation in highly diverse plots as described in Chao et al.⁶⁶ using the iNext⁶⁷ package in R. We then compared the results to the traditional Simpson index computed above and obtained similar results (see Supplementary Fig. 1). Therefore we conducted further analysis with the traditional Simpson diversity index.'

We also clearly state as suggested by the reviewer that we use a null model to standardize phylogenetic diversity. We do this as follows:
'We used a null model based on frequency, which randomized community data abundances within species, while maintaining the same species occurrence frequency.'

Table 3. Comparison of already calculated Simpson index and the newly calculated iNEXT Simpson index as Hill numbers, both are highly correlated.

Plot	Simpson	iNext Hill Simpson
ASN_02	-0.0010	-0.399
ASN_04	0.0010	2.284
BBR_02	0.0030	1.964
BBR_14	-0.0040	-1.916
BBR_16	-0.0010	-1.794
BBR_17	-0.0120	-4.386
BOR_05	-0.0010	-1.702
BOR_06	0.0050	4.327
CAP_09	-0.0040	-1.714
CAP_10	0.0130	17.746
DAD_03	0.0170	12.996
DAD_04	0.0080	4.292
DRA_04	0.0030	1.993
DRA_05	-0.0020	-1.916
ESU_18	0.0100	4.418
FUR_07	0.0020	0.819
FUR_08	0.0080	5.095
KDE_01	-0.0050	-1.367
KDE_02	0.0060	7.117
TON_01	-0.0050	-0.541
TON_08	0.0050	3.286

Figure 3. Posterior distribution test re-done with the original Simpson index and the new Simpson as Hill numbers from iNEXT. Both indices were standardised to account for differences in time between censuses. Both results show the same pattern of results supporting our findings of stronger decreases in taxonomic diversity in drier forests (see also Table 4 below).

	mean	median	mode	HDI%	HDI-low	HDI-up
Difference	-0.19454	-0.1938	-0.2035	89	-0.37	-0.011
Sigma difference	-0.14501	-0.14107	-0.13056	89	-0.31	0.009

HDI= Highest density interval

R2-3: Authors expect “soils with higher water holding capacity may buffer drought impacts on forest communities”. “forest soils high in clay may be able to maintain higher water availability over longer periods during droughts than sandy soils where the water holding capacity tends to be lower” (Line 61-62). “PC3 was mainly loaded by the soil texture characteristics as percentage of Clay and Sand and is thus referred to as a soil texture axis.” (Lines 429-430). However, they found that PC1 (mainly representing cation exchange capacity, exchangeable magnesium (Mgex) and nitrogen content) were correlated with phylogenetic diversity change. I am not familiar with how soil water holding capacity relates to soil nutrients. I guess there are some missing link here.

Response:

We thank the reviewer for highlighting there was something missing in our expectations about the effects of soil nutrients and water availability on the possible changes in the three diversity facets. We have updated the section on hypothesis and expectations in our manuscript to link the role that nutrients may play on forest distributions with a drying climate. The new text is as follows:

‘Responses in the three diversity facets may be determined by soil characteristics in addition to climatic conditions; for example, soils rich in nutrients and with higher water holding capacity may buffer drought impacts on forest communities⁷, as drought resilience may vary not only with depth to water table but also with soil nutrient content²⁵. Therefore, we also investigate the role of soil characteristics on the response of the three diversity facets along the climatic gradient and across time. Our results fill knowledge gaps on the coordination of changes in biodiversity in tropical forest as a response to climate changes, and on the extent to which forest communities may be susceptible to a changing environment depending on their current position along the climatic gradient.’

Reviewers' comments second round:

Reviewer #2 (Remarks to the Author):

Thanks authors for having very good response for my comments. I have no further comments now.

Reviewer #3 (Remarks to the Author):

Thank you for giving me an opportunity to review the manuscript titled "Long-term droughts may drive drier tropical forests towards increased functional, taxonomic and phylogenetic homogeneity by Aguirre-Gutiérrez et al. I am not an expert in forest dynamics however I do have experience evaluating functional, taxonomic, and phylogenetic trends in fish communities. Further, I have extensive experience in Bayesian inference including applying Bayesian methods to ecological systems outside my immediate area of expertise, fisheries ecology. Thus, I will restrict my comments to the statistical procedures in the manuscript.

The research attempts to evaluate functional, taxonomic, and phylogenetic trends of vegetation in a forest located in West Africa. There are two primary analyses. The first compares diversity indexes between dry and wet tropical forests. The second is a series of models that evaluate the influence of climate and soil variables. The first analysis is rather straight forward and applies a simple Bayesian version of a t-test as implemented in the BEST package. However, I do have a comment on the choice of priors that I will elaborate on below. The second approach is more complex and uses a series of NHST correlations and multivariate techniques to distil potential explanatory variables to a subset (or representation of the data = PCA axes) and those variables are subjected to a Bayesian model selection procedure (LOO). As mentioned in a previous review of this manuscript, model selection can be a tricky endeavor and I agree. Nevertheless, model selection is often necessary to identify the most important explanatory variables in the model. I have serious concerns about how the authors treated the data to prepare it for model selection and the model selection itself. Unfortunately, the issues presented below render the modeling exercise incorrect and any inference drawn meaningless. However, I could be misreading the text or important information was left out. I will attempt to outline what I see as serious flaws with the statistical analysis. I will also add, these issues could be fixed but not without significant effort that would likely change the results and interpretations, thus producing a manuscript that is much different than what has been submitted.

Plot area and time between census correlations:

My first concern is the use of NHST to evaluate the relationship between plot area and time between census with the three diversity indices. Pearson's correlation analysis assumes normality of data, an assumption that is not likely met. Area is negatively skewed however time could be considered normal, at least has minimal skew (figure based on Supplementary Table 1 attached). There are other analyses that do not require normality. However, my main concern here is not which analysis is used to determine a correlation, it is that an analysis is used to determine the relationship between an independent variable and dependent variable before modeling. If there are biological reasons to include these variables as potential predictors (I think there is) then they should have been included in the model selection procedure. Further, using a NHST cutoff ($\alpha = 0.05$) to declare a biological significance should not be conducted (which was done when the authors decided some variables were not important to include in the model selection), particularly when using Bayesian analysis. Even if the authors used maximum likelihood for their model selection (e.g., AIC) the initial application of p-values to evaluate what variables should be included is incorrect. Burnham and Anderson (2003) note that variables that have a p-value > 0.05 can still have predictive information and thus could be selected in the "best" model with using AIC. I agree with the previous reviewer that plot area and time between census could influence the outcomes but they should not have been evaluated before model selection using p-values.

Principal components analysis of soil data:

The flaw in the way PCA axes were used renders all results incorrect. First, very little information is provided on how the PCA was conducted. Why were only three axes selected? There are several cutoffs typically used (e.g., % variance explained > 10) but no indication. The authors do mention only using three axes to prevent model overfitting, however, that can be handled with LOO. Using fewer PCA axes can omit valuable information that is ignored without biological reason. For example, C(%), N(%), and NA are all positively loaded on each axes. An argument could be made that the highest loadings would identify the axes that represents the trend in these variables but it would be difficult. It is possible these variables are being described by the fourth PCA axis. Without a transparent way of selecting axes it is difficult to determine. At best, the PCA needs to be better explained in the methods and a scatterplot of the axes should be included in the appendix to help the reader interpret the results. A scatterplot would also help with my second concern.

PCA axis values generally range from some negative value to some positive value. Sites that have a large negative value are considered much different than sites with large positive values. Further, each axis represents a gradient of some component. Say for example axis 1 represented a gradient of Sand(%) and Fe where large negative values of axis 1 represent high percent of sand and high concentrations of Fe and large positive values for axis 1 would then represent low percent of sand and low concentrations of Fe. Here, a site that has a large negative value for axis 1 (say -3) would thus be characterized as having high percent of sand and high concentrations of Fe. Now compare that to a site that has a large positive value (say +3) which would be characterized as having low percent of sand and low concentration of Fe. The next step is important, the authors report using a quadratic term for all PCA axes (Supplementary Table 4 and line 455 of main document). By taking the square of a PCA axis the large negative value and large positive values would be the same. And in the example above, both sites would be 4 (-2^2 and 2^2) meaning they are the same but they are in fact extremely different based on the PCA results.

Models:

It is not clear why interactions terms were used as potential models without their main effects. Yes, it is possible to have a reason for using only the interaction terms without their main effects but it is uncommon and I do not see an explanation to support this decision. The problem here is that by omitting the main effects the model assumes the expected mean response when the two predictor variables in the interaction are zero is the same. In other words model 14 ($PC1 * \Delta MCWDAbs$) assumes the diversity index is the same for observations with $PC1=0$ and $\Delta MCWDAbs=0$ but the slopes are then different resulting in the predicted lines to always intersect at 0.

Use of HDI 89%:

The use of 89% HDI is unconventional. First, I could not find anywhere in the Kruschke reference that recommends using 89%, he uses 95% throughout. I don't think the authors should use 95% (the value that typically is used) blindly but whatever range is selected should be selected for a reason. I think the authors used 89% because it is the default range used by the bayestestR package. According to the directions for bayestestR 89% is used as a default because it is "the highest prime number that does not exceed the already unstable 95% threshold. What does it have to do with anything? Nothing, but it reminds us of the total arbitrariness of any of these conventions (McElreath, 2018)." The authors should remove the reference to Kruschke and provide justification for using whatever range for their HDI as it appears the default 89% was selected in jest.

Use of pooled mean and sd for priors:

I am aware the Kruschke's paper on BEST uses the pooled mean and pool sd (times a large number) however this is a poor decision in practice. The prior used in the paper is likely flat (given the large sd) and the posterior is most likely influenced more by the data. But using the data to inform the prior and inform the model should never be done. I do not think using a traditional flat prior (normal with mean = 0 and SD = 10) will change the results but it should be used instead of the data to inform the prior. Unless the authors can provide justification for doing so.

Long-term droughts may drive drier tropical forests towards increased functional, taxonomic and phylogenetic homogeneity

Jesús Aguirre-Gutiérrez, Yadvinder Malhi, Stephen Adu-Bredu, Kofi Affum-Baffoe, Timothy R. Baker, Sophie Fauset, Agne Gvozdevaite, Wannes Hubau, Simon L. Lewis, Sam Moore, Theresa Peprah, Kasia Ziemińska, Oliver L. Phillips, Imma Oliveras

Response to Comments from Referee 3

We thank the reviewer for taking the time to review our manuscript and for the insightful comments. Below we respond to the comments given and explain what we have done in the main manuscript in regards to new analyses, new text, figures and tables. In summary we have re-run our analyses following the advice and updated all material when needed.

1) Plot area and time between census correlations:

My first concern is the use of NHST to evaluate the relationship between plot area and time between census with the three diversity indices. Pearson's correlation analysis assumes normality of data, an assumption that is not likely met. Area is negatively skewed however time could be considered normal, at least has minimal skew (figure based on Supplementary Table 1 attached). There are other analyses that do not require normality. However, my main concern here is not which analysis is used to determine a correlation, it is that an analysis is used to determine the relationship between an independent variable and dependent variable before modeling. If there are biological reasons to include these variables as potential predictors (I think there is) then they should have been included in the model selection procedure. Further, using a NHST cutoff ($\alpha = 0.05$) to declare a biological significance should not be conducted (which was done when the authors decided some variables were not important to include in the model selection), particularly when using Bayesian analysis. Even if the authors used maximum likelihood for their model selection (e.g., AIC) the initial application of p-values to evaluate what variables should be included is incorrect. Burnham and Anderson (2003) note that variables that have a p-value > 0.05 can still have predictive information and thus could be selected in the "best" model with using AIC. I agree with the previous reviewer that plot area and time between census could influence the outcomes but they should not have been evaluated before model selection using p-values.

R1- Following the recommendation of the reviewer we now account for the time between censuses and the area of the plot in our modelling protocol directly and have modified the text accordingly. We do not test their relation with the diversity facets before the modelling protocol. We model the yearly rate of change in the diversity facets of our forests ecosystems (which accounts for the differences in time between census) and include the plot area as a covariate in our models. Our results are now updated and we show that the important parameters continue being the same for the functional and taxonomic diversity as before. The models of phylogenetic diversity improved and render more support to our hypothesis of the relationship between water availability and soil conditions for determining phylogenetic diversity changes (Lines 73-77).

Our new text is specified in lines 438 to 442:

"Subsequently, we modelled the observed rate of changes in each of the three diversity facets ($\Delta FDis_r$, $\Delta Simpson_r$ and ΔMPD_r) as a function of the climatic variables specified above and soil characteristics (three first PCA axes). As some plots were smaller than 1 ha (Supplementary Table 1) we included plot size as a covariate in the statistical models to account for its possible effect in the observed changes in the three diversity facets."

2) Principal components analysis of soil data:

2.1-The flaw in the way PCA axes were used renders all results incorrect. First, very little information is provided on how the PCA was conducted. Why were only three axes selected? There are several cutoffs typically used (e.g., % variance explained > 10) but no indication. The authors do mention only using three axes to prevent model overfitting, however, that can be handled with LOO. Using fewer PCA axes can omit valuable information that is ignored without biological reason. For example, C(%), N(%), and NA are all positively loaded on each axes. An argument could be made that the highest loadings would identify the axes that represents the trend in these variables but it would be difficult. It is possible these variables are being described by the fourth PCA axis. Without a transparent way of selecting axes it is difficult to determine. At best, the PCA needs to be better explained in the methods and a scatterplot of the axes should be included in the appendix to help the reader interpret the results. A scatterplot would also help with my second concern.

R2.1- We selected the number of PCA axes (see Figure 1 below) by means of the contribution to variance explained. In our PCA analysis the first three PCA axes explain each more than 10% of the variance (Figure 1a-c), the three together explain most variance in the data (76.2%) and axis 4 and onwards explain very few of such variance (Figure 1d). Therefore we decided to selected only the first three axis (Figure a-c). We agree that more clarity about the selection is needed and we have updated the text as follows and have included Figure 1 in the supplementary information of our manuscript (Supplementary Figure 7). Our new text focusing on the PCA axes selection is as follows in lines 410-412:

“We used the first three principal component axes as they explain at least 10% of the variance, the three together explain most variance in the data (76.2%) and axis four and onwards explain >10% of data variance (Supplementary Figure 7).”

Figure 1. Results of the Principal Component Analysis from which the first three axis (a-c) were selected and d) variance explained by each axis.

2.2-PCA axis values generally range from some negative value to some positive value. Sites that have a large negative value are considered much different than sites with large positive values. Further, each axis represents a gradient of some component. Say for example axis 1 represented a gradient of Sand(%) and Fe where large negative values of axis 1 represent high percent of sand and high concentrations of Fe and large positive values for axis 1 would then represent low percent of sand and low concentrations of Fe. Here, a site that has a large negative value for axis 1 (say -3) would thus be characterized as having high percent of sand and high concentrations of Fe. Now compare that to a site that has a large positive value (say +3) which would be characterized as having low percent of sand and low concentration of Fe. The next step is important, the authors report using a quadratic term for all PCA axes (Supplementary Table 4 and line 455 of main document). By taking the square of a PCA axis the large negative value and large

positive values would be the same. And in the example above, both sites would be 4 (-2^2 and 2^2) meaning they are the same but they are in fact extremely different based on the PCA results.

R2.2- We agree in the point raised by the reviewer about the effect of quadratic terms of PCA axes in the modelling protocol. Therefore, we have re-run all models including only linear terms and have updated our model specifications in Supplementary Table S4. Our results continue supporting our original findings and provide more evidence now for our hypothesised combined effect of soil and climate on the effect of the phylogenetic diversity changes.

3) Models:

It is not clear why interaction terms were used as potential models without their main effects. Yes, it is possible to have a reason for using only the interaction terms without their main effects but it is uncommon and I do not see an explanation to support this decision. The problem here is that by omitting the main effects the model assumes the expected mean response when the two predictor variables in the interaction are zero is the same. In other words model 14 ($PC1 * \Delta MCWD_{Abs}$) assumes the diversity index is the same for observations with $PC1=0$ and $\Delta MCWD_{Abs}=0$ but the slopes are then different resulting in the predicted lines to always intersect at 0.

R3- All models generated included the main effects. We agree this was not clear in our original Supplementary Table S4 where we tried to summarise the models used. We have now improved our table and specify all terms as used in each model. See Supplementary Table S4.

R1-2-3. Resulting models after accounting for concerns 1-2-3.

Here (Figure 2 below) we show the new results after following the advice from the reviewer. We show how the models for changes in functional and taxonomic diversity did not change in essence. Models for phylogenetic diversity now support more our initial hypothesis of interacting effect of climate and soil for determining changes in diversity facets (Lines 73-77). The new results are shown in Table 1, Supplementary Table 6 and Figure 2 and have also been updated in lines 140-157 and are included in the discussion section in lines 182-212.

Figure 2. Climatic and soil drivers of observed rates of change in a) functional (ΔFD_{Is}), b) taxonomic ($\Delta Simpson$) and c-h) phylogenetic (ΔMPD) diversity in West African forest communities. Changes in functional and taxonomic diversity were mainly explained by the absolute changes in the maximum climatic water deficit ($\Delta MCWD_{Abs}$). Observed changes in phylogenetic diversity were better explained by the soil characteristics covered by the three PC axes (Supplementary Table 3) and their interaction with climatic drivers ($\Delta MCWD_{Abs}$, $\Delta MCWDFull$, $\Delta VPDAbs$). PC1: eCEC(+), magnesium(+) and nitrogen(+); PC2: pH(-), Fe(+) and Ca(-); PC3: %Clay(-) and %Sand(+). The solid black fitted line shows the mean posterior prediction for the functional and taxonomic diversity change models. The red and blue fitted lines show the mean posterior predictions for the phylogenetic diversity based on the minimum and maximum values of the climatic drivers included in the model (Table 1). Grey lines show 700 random draws from the posterior distribution representing variability of the expected model fit.

4) Use of HDI 89%:

The use of 89% HDI is unconventional. First, I could not find anywhere in the Kruschke reference that recommends using 89%, he uses 95% throughout. I don't think the authors should use 95% (the value that typically is used) blindly but whatever range is selected should be selected for a reason. I think the authors used 89% because it is the default range used by the bayestestR package. According to the directions for bayestestR 89% is used as a default because it is "the highest prime number that does not exceed the already unstable 95% threshold. What does it have to do with anything? Nothing, but it reminds us of the total arbitrariness of any of these conventions (McElreath, 2018)." The authors should remove the reference to Kruschke and provide justification for using whatever range for their HDI as it appears the default 89% was selected in jest.

R4- We have now included the right citation (Makowski et al. 2019) and give a justification for its use as suggested by Makowski et al. 2019. The selection of the 95% HDI would just follow the protocol from the frequentist approach. We selected the 89% HDI instead of the 95% following Makowski et al. (2019), where it is argued that HDI below 95% would be more stable and more appropriate when less than 10000 posterior samples are drawn. They suggest using the 89% HDI and we followed this advice. We now include the above explanation in the main text of the manuscript in lines 455-4457:

"We computed the highest density intervals (HDI) rendering the range containing the 89% most probable effect values as suggested in Makowski et al.⁸¹ as this may result in more stable predictions when less than 10000 posterior samples are drawn."

Makowski, D., Ben-Shachar, M. S., & Lüdtke, D. (2019). bayestestR: Describing Effects and their Uncertainty, Existence and Significance within the Bayesian Framework. Journal of Open Source Software, 4(40), 1541. <https://doi.org/10.21105/joss.01541>

5) Use of pooled mean and sd for priors:

I am aware the Kruschke's paper on BEST uses the pooled mean and pool sd (times a large number) however this is a poor decision in practice. The prior used in the paper is likely flat (given the large sd) and the posterior is most likely influenced more by the data. But using the data to inform the prior and inform the model should never be done. I do not think using a traditional flat prior (normal with mean = 0 and SD = 10) will change the results but it should be used instead of the data to inform the prior. Unless the authors can provide justification for doing so.

R5- As the reviewer suggest modifying the prior to a flat prior as stated above is unlikely to modify the results. We corroborated this by re-running the test with the BEST package and specifying the priors suggested (mean=0 and SD=10). The results of this new analysis are virtually the same than the original ones and can be seen in the figure below (Figure 3). We follow the advice of the reviewer and present the new results in our main Figure 2 (d-f) and lines 106-118 in main text. We have also modified the methods about this analysis to update the values of the priors used as suggested in lines 429-434.

Figure 3. Rate of change ($\Delta FDIs_r$, $\Delta Simpson_r$, and ΔMPD_r) in the three diversity facets after grouping the vegetation plots as belonging to dry or wet forest. The horizontal dotted line represents no change in diversity with positive values showing increases and negatives values decreases in diversity. The insets on the bottom right corner show the average difference in diversity change between dry and wet forests. Negative average difference values depict a stronger loss in the diversity facet in drier forests in comparison to wetter forests. The posterior highest density intervals (HDI-l: lower; HDI-u: upper) and probability change (Prob) values are also shown.

Reviewer #3's comments third round:

The authors have addressed the concerns raised in my previous review. However one concern remains and I have one comment:

Should Table 1 metrics be changed to the rate of change for each response?

Use of HDI 89%: The authors have updated the reference, however, the reference is to the R package which does not justify 89% and my concerns remain. The R package (bayestestR) documentation states that 89 indicates the arbitrariness of interval limits and its only remarkable property is it being the highest prime number that does not exceed 95%. The referenced article (Makowski et al. 2019) also references Kruschke (2015) for needing an effective sample size, NOT posterior samples as indicated by the authors, of at least 10,000 if 95% intervals are used. Kruschke (2015) does not suggest 89% is a better alternative. Why not use 90% instead of 89%? Pages 184-186 in Kruschke (2015) discusses this heuristic and states this recommendation is based on experience and not a requirement. Further, Kruschke indicates this heuristic should be followed for "reasonable stability" in the estimates of the limits. Neither of the references present evidence that 89% is any better than 95% at providing a more stable range of CI. Rather, Kruschke demonstrates that 95%CI are more stable when $ESS \geq 10000$. Makowski et al. (2019) "deems" 89% to be more stable but provides no evidence and does not present an argument for selecting 89%. My recommendation is to select a posterior summary that best describes the distribution. I would suggest several different quantiles, this can help describe to the reader how skewed the data are. Finally, it would be a valid statement to say 95% was not used because this range has been shown to be unstable with $ESS < 10000$ then cite Kruschke (2015). It is not valid to suggest 89% is more stable than 95%. Also, please specify effective sample size, NOT posterior samples (they are two different things). I do not think this is a major problem as long as the references and selection of the summary is accurately described. Most importantly, the way it is written, the authors are saying the MCMC chain just needs 10000 draws for stable predictions, which is not true.

Response to Reviewer Comments fourth round:

Reviewer 1.1: The authors have addressed the concerns raised in my previous review. However one concern remains and I have one comment: Should Table 1 metrics be changed to the rate of change for each response?

Response: R1- We have updated Table 1 as requested.

Reviewer 1.2: Use of HDI 89%: The authors have updated the reference, however, the reference is to the R package which does not justify 89% and my concerns remain. The R package (bayestestR) documentation states that 89 indicates the arbitrariness of interval limits and its only remarkable property is it being the highest prime number that does not exceed 95%. The referenced article (Makowski et al. 2019) also references Kruschke (2015) for needing an effective sample size, NOT posterior samples as indicated by the authors, of at least 10,000 if 95% intervals are used.

Kruschke (2015) does not suggest 89% is a better alternative. Why not use 90% instead of 89%? Pages 184-186 in Kruschke (2015) discusses this heuristic and states this recommendation is based on experience and not a requirement. Further, Kruschke indicates this heuristic should be followed for "reasonable stability" in the estimates of the limits. Neither of the references present evidence that 89% is any better than 95% at providing a more stable range of CI. Rather, Kruschke demonstrates that 95%CI are more stable when $ESS \geq 10000$. Makowski et al. (2019) "deems" 89% to be more stable but provides no evidence and does not present an argument for selecting 89%.

My recommendation is to select a posterior summary that best describes the distribution. I would suggest several different quantiles, this can help describe to the reader how skewed the data are. Finally, it would be a valid statement to say 95% was not used because this range has been shown to be unstable with $ESS < 10000$ then cite Kruschke (2015). It is not valid to suggest 89% is more stable than 95%. Also, please specify effective sample size, NOT posterior samples (they are two different things). I do not think this is a major problem as long as the references and selection of the summary is accurately described. Most importantly, the way it is written, the authors are saying the MCMC chain just needs 10000 draws for stable predictions, which is not true.

Response: R1- We have updated Table 1 as requested.

Following the recommendation from the reviewer we have included the 50%, 89% and 95% HDI to summarise the data in Table 1. We have also updated our manuscript as to include the text suggested by the reviewer. The text in lines 455-459 is as follows: 'We computed the highest density intervals (HDI) rendering the range containing the 89% most probable effect values as suggested in Makowski et al.⁸⁷ and calculated the ROPE values using such HDI. Although the 95% HDI was not used as this range has been shown to be unstable with $ESS < 10000$ (effective sample size)³² we also present it together with the 50% HDI as to give a more complete description of the data.'